# Enhanced JunD/RSK3 signalling due to loss of BRD4/FOXD3/miR-548d-3p axis determines BET inhibition resistance

Fang Tai[1,2,3], Kunxiang Gong[1,2,3], Kai Song[1,2,3], Yanling He[1,2,3] & Jian Shi[1,2,3]*

BET bromodomain inhibitors (BETi), such as JQ1, have been demonstrated to effectively kill multiple types of cancer cells. However, the underlying mechanisms for BETi resistance remain largely unknown. Our evidences show that JQ1 treatment evicts BRD4 from the FOXD3-localized MIR548D1 gene promoter, leading to repression of miR-548d-3p. The loss of miRNA restores JunD expression and subsequent JunD-dependent transcription of RPS6KA2 gene. ERK1/2/5 kinases phosphorylate RSK3 (RPS6KA2), resulting in the enrichment of activated RSK3 and blockade of JQ1 killing effect. Dual inhibition of MEKs/ERKs or single EGFR inhibition are able to mimic the effect of JunD/RSK3-knockdown to reverse BETi resistance. Collectively, our study indicates that loss of BRD4/FOXD3/miR-548d-3p axis enhances JunD/RSK3 signalling and determines BET inhibition resistance, which can be reversed by targeting EGFR-MEK1/2/5-ERK1/2/5 signalling.

[1] Department of Pathology, Nanfang Hospital, Southern Medical University, Guangzhou 510515 Guangdong, China. [2] Department of Pathology, School of Basic Medical Science, Southern Medical University, Guangzhou 510515 Guangdong, China. [3] Guangdong Provincial Key Laboratory of Molecular Tumor Pathology, Southern Medical University, Guangzhou 510515 Guangdong, China. *email: jianshismu@126.com

Despite years of treatment, the 5-year survival rates of patients with basal-like breast cancer (BLBC) remain <30%. BLBC is insensitive to endocrine or HER2-targeted therapies compared with other breast cancer subtypes and is prone to distant metastasis in early-stage disease[1,2]. Owing to the lack of targeted therapy, chemotherapy is still the primary option for treatment of BLBC, and the pathological complete remission rate of BLBC with chemotherapy is 50%. However, patients who are resistant to chemotherapy soon decline[3]. Therefore, the development of targeted therapies is critical for the treatment of BLBC. In recent years, although several potential molecular targets of BLBC have been identified[4–6], effective targeted strategies for clinical treatment of BLBC patients are still not available.

BET family chromatin-binding proteins contain two tandem N-terminal bromodomains and an extra-terminal domain. One typical BET family members, BRD4, is tightly related to the transcriptional activation by remaining associated with chromatin through recognition of acetylated histone proteins[7]. Initially, BRD4 was identified to mark the transcriptional start sites of G1 phase related genes, accelerate their expression and promote cell-cycle progression to S phase[8,9]. As a protein scaffold, BRD4 interacts with a variety of proteins on chromatin. For example, BRD4 is one of the proteins that recruit positive transcriptional elongation factor complex to RNA polymerase II at the active gene promoter; and can also co-localize with several transcription factors at the specific active transcriptional sites[10]. Recently, BRD4 was shown to preferentially localize at the super-enhancers of a series of critical oncogenes, initiating and maintaining their expression in tumour cells[11]. These findings indicate that BRD4 is a critical driver of oncogene expression which tumour cells depend on for their survival and proliferation. Currently, BET inhibitors, small-molecule compounds targeting bromodomain, are newly emerging agents for therapeutic strategies for cancer, and have been used in several clinical trials for cancer, showing encouraging results, especially in acute myeloid leukaemia[12]. Our previous study observed that BET inhibitors, JQ1 and MS417, significantly suppress the cancer stem cell-like properties and tumorigenicity of BLBC cells[5]. Unfortunately, similar to other targeted therapy drugs, the anti-tumour efficacy of BET inhibitors is challenged by the intrinsic or adaptive drug resistance of cancer cells[13,14]. Therefore, the discovery of the underlying resistance mechanisms is pivotal to optimizing the clinical efficacy of these drugs. Recent studies indicated that the maintenance of the protein levels of BRD4 or its bromodomain independent function in tumour cells contributes to BET resistance[15–19]. However, the potential compensatory survival molecules activated upon BET inhibition are barely known.

In this study, we identify compensatory JunD/RSK3 survival signalling resulting from the loss of the BRD4/FOXD3/miR-548d-3p axis upon BET inhibition and develop a dual BRD4/EGFR inhibition strategy that inhibits BET protein and derivative survival signals concomitantly for overcoming BETi resistance.

## Results

**Elevated RSK3 expression is responsible for BETi resistance.** To reveal compensatory survival-related molecules upon BET inhibition in BLBC cells, we incubated MDA-MB-231, a typical BLBC cell line, with JQ1 (1 μM), the first BET inhibitor[20], for 24 h, extracted its total RNA and analysed the transcriptomes of control and JQ1-treated cells by RNA sequencing (GSE140003). Because kinases are relatively druggable targets compared with other types of molecules, the genes that encoded kinases, that were significantly upregulated in JQ1-treated cells compared with control cells, were selected and considered as potential drug resistance genes. The most notable inducible

kinase gene was *RPS6KA2* (Supplementary Fig. 1A), which encodes RSK3, a member of the p90 ribosomal S6 kinase family. RSKs are directly phosphorylated and activated by MEK/ERK signalling, which are involved in transcription, translation, and cell-cycle regulation[21–24]. However, the pathological role of RSK3 in BLBC and its transcriptional regulation remain unclear. Consistent with the RNA sequencing data, the protein and mRNA expression of RSK3 were significantly induced by JQ1 (1 μM) treatment within 24 h in BLBC cell lines, MDA-MB-231 and BT549 (Fig. 1a and Supplementary Fig. 1B).

To discern the pathological significance of inducible RSK3 in BLBC cells upon BET inhibition, stable clones of RSK3-overexpressing BLBC cells were generated to mimic the above situation (Supplementary Fig. 1C). We treated the vector control and overexpressing clones with JQ1 (1 μM) for 48 h, and observed that the upregulation of this gene obviously compromised the JQ1-mediated killing effect detected by the CellTiter-Glo® luminescent cell viability assay (Fig. 1b). Similar results were observed in that overexpression of RSK3 partially reversed JQ1-mediated suppression of tumoursphere formation (Fig. 1c and Supplementary Fig. 1D) and significantly ameliorated JQ1-induced apoptosis (Fig. 1d). These data suggest that the induction of RSK3 is not required for JQ1-mediated cell growth arrest and apoptosis, but might be responsible for drug resistance. Next, we constructed stable clones of *RPS6KA2*-knockdown in BLBC cells (Supplementary Fig. 1E). Intriguingly, compared with the vector control cells, *RPS6KA2*-knockdown clones were more sensitive to JQ1 (Fig. 1e), and silencing of *RPS6KA2* also greatly enhanced the JQ1-induced apoptosis (Fig. 1f) and promoted the JQ1-mediated inhibition of tumoursphere formation (Fig. 1g and Supplementary Fig. 1F).

Furthermore, we sought to analyse the tumourigenic potential of vector control and *RPS6KA2*-knockdown cells as well as their response to JQ1 in a xenograft model. To this end, we injected BALB/c nude mice with *RPS6KA2*-knockdown MDA-MB-231 cells or empty vector. JQ1 treatment at 35 mg/kg was started when tumours reached an average volume of 90 mm[3]. *RPS6KA2*-knockdown cells exhibited proliferation rates similar to those of the control cells in vehicle-treated mice, but they were more sensitive to JQ1-mediated growth-inhibitory and killing effect than vector control cells (Fig. 1h, i and Supplementary Fig. 1G). The data showed that *RPS6KA2* acts as an inducible resistance gene upon BET inhibition in BLBC cells.

**JunD-dependent *RPS6KA2* transcription mediates BETi resistance.** Next, we sought to explore the mechanism of the emergent induction of RSK3. Based on the RNA sequencing data, the expression of JunD was rapidly stimulated by JQ1 within 24 h that was confirmed by protein analysis (Fig. 2a). Interestingly, by searching the enhancer region of *RPS6KA2* gene, we found a potential JunD binding site, GTGACTCT (−2161 bp upstream of the translation start site) (Fig. 2b). ChIP data revealed that this region contains strong H3K4me1 signals (Supplementary Fig. 2A). JunD, a member of the activator protein-1 (AP-1) family, is a powerful transcription factor that can regulate apoptosis and protect against oxidative stress by modulating the genes involved in antioxidant defence and hydrogen peroxide production[25]. To study whether JunD is responsible for the direct induction of *RPS6KA2* transcription, a wild-type *RPS6KA2* gene enhancer luciferase reporter was constructed by inserting this 2000 base-pair fragment, and the potential JunD recognition motif in the enhancer was mutated (Fig. 2b). Luciferase experiments in MDA-MB-231 and BT549 cells showed that JQ1 (1 μM) treatment for 6 h apparently enhanced the luciferase reporter activity by nearly four-fold, while knockdown of JunD

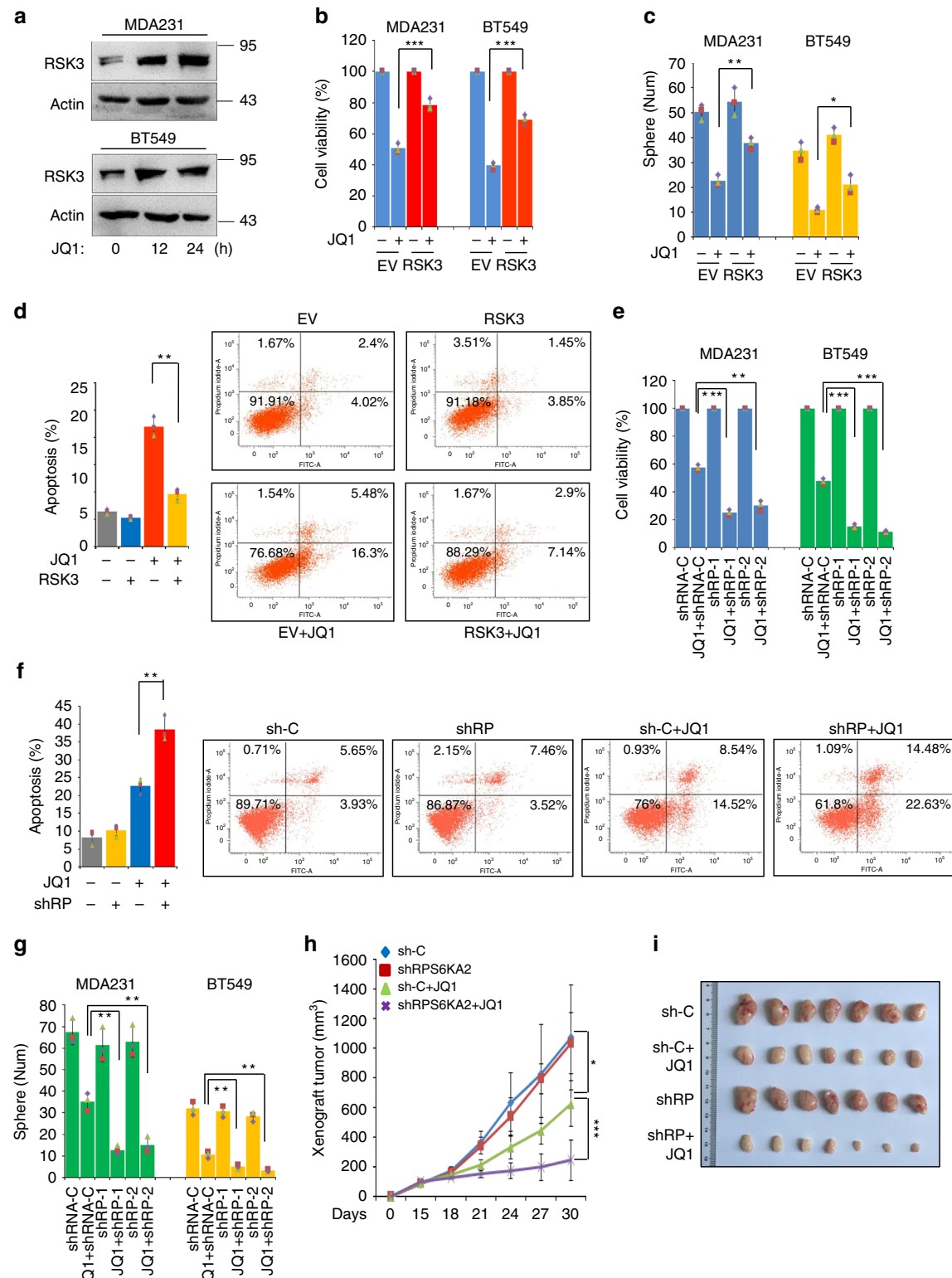

significantly abolished the induction of luciferase activity (Fig. 2c). Similar results were observed in luciferase reporter transfected HEK293 cells upon JQ1 treatment; ectopic JunD expression obviously stimulated the luciferase activity and enhanced the effect of JQ1. Moreover, mutation of the potential JunD binding site inhibited JQ1 and JunD induced luciferase activity (Fig. 2d). Next, chromatin immunoprecipitation (ChIP)-qPCR assay was performed to determine whether JunD directly binds to the *RPS6KA2* gene enhancer. Results from MDA-MB-

231 and BT549 cells showed that JQ1 treatment for 6 h strongly stimulated the occupancy of JunD protein on the *RPS6KA2* gene enhancer, which was ameliorated by knockdown of JunD (Fig. 2e), indicating that JunD directly activates the *RPS6KA2* gene transcription. Similar results were obtained by EMSA assay (Supplementary Fig. 2B). At the same time, we detected the binding status of c-Jun, JunB and c-Fos compared with that of JunD. Interestingly, all four proteins recognized the *RPS6KA2* enhancer in the absence of JQ1 treatment; c-Jun and JunD had

**Fig. 1 Elevated RSK3 is responsible for BETi resistance. a** Western blotting was performed to detect the protein levels of RSK3 in MDA-MB-231 and BT549 cells treated with DMSO or JQ1 (1 μM) for 0, 12 and 24 h. **b** The vector controls and RSK3-overexpressing BLBC cell clones were treated with DMSO or JQ1 (1 μM) for 48 h, and luminescent cell viability assays were performed to measure the killing effects. Statistical data (mean ± SD) are shown (***$P < 0.001$, one-way ANOVA). **c** Tumoursphere formation in RSK3-overexpressing BLBC cells and their vector controls was observed with or without JQ1 (1 μM) treatment. Statistical data of tumoursphere numbers are shown (*$P < 0.05$, **$P < 0.01$; one-way ANOVA). **d** Measurement of apoptosis in RSK3-overexpressing MDA-MB-231 cells and vector control with or without JQ1 (1 μM) treatment by Annexin V-FITC flow cytometry assay. Statistical data (mean ± SD) are shown based on three independent experiments. A representative experiment is shown (**$P < 0.01$, one-way ANOVA). **e** RPS6KA2-knockdown clones and shRNA control BLBC cells were treated with JQ1 (1 μM) for 48 h, the killing effects were detected by luminescent cell viability assay (**$P < 0.01$, ***$P < 0.001$; one-way ANOVA). **f** Annexin V-FITC flow cytometry assay was used to measure the synergistic pro-apoptotic effects of JQ1 on the silencing of RPS6KA2. Statistical data of three independent experiments are shown. A representative experiment is shown (**$P < 0.01$, one-way ANOVA). **g** Tumoursphere was counted in RPS6KA2-knockdown BLBC cells and their shRNA controls in the absence or presence of JQ1 (1 μM) treatment. Statistical data (mean ± SD) were shown (**$P < 0.01$, one-way ANOVA). **h–i** BALB/c nude mice were injected with RPS6KA2-knockdown or shRNA control MDA-MB-231 cells ($n = 7$), following treatment with vehicle control or JQ1 (35 mg/kg). After 25 days, tumours were weighed and images were taken. **h** Growth curves of xenograft tumours are shown (*$P < 0.05$, ***$P < 0.001$); **i** Photographs of tumours are shown. Source data are provided as a Source Data file.

the stronger binding affinity, while JunB and c-Fos showed a much weaker association. Upon JQ1 treatment, the binding of c-Jun was significantly decreased; although the association of JunB and c-Fos was slightly elevated. However, the binding affinity of JunD on RPS6KA2 enhancer was robustly enhanced in the presence of JQ1 (Supplementary Fig. 2C). Taken together, we reason that JunD is most likely to determine the responsive RPS6KA2 expression and BETi resistance.

Further, JUND was stably knocked down in MDA-MB-231 and BT549 cells to clarify its role in the induction of RSK3 and BETi resistance. Silencing of JUND repressed JQ1-induced RSK3 expression in BLBC cells (Fig. 2f), and re-sensitized these cells to JQ1, while the functional consequence of silencing of JUND was similar to that of RPS6KA2-knockdown (Fig. 2g). Consistently, in contrast to vector control cells, the JQ1-mediated suppressive effect on tumoursphere formation was much stronger in JUND-knockdown cells (Fig. 2h and Supplementary Fig. 2D). Rescue expression of RSK3 in the JUND-knockdown BLBC clones restored the resistance to JQ1 (Fig. 2i), directly reflecting the fact that these two molecules function interdependently.

We also generated the JunD-overexpressing stable clones in BLBC cell lines. In line with our speculation, the expression of JunD significantly enhanced JQ1-induced RSK3 expression (Fig. 2j), and relieved the cytotoxic effect of JQ1 (Fig. 2k). Similarly, JunD overexpression partially inhibited the JQ1-mediated suppressive effect on tumoursphere formation ability (Supplementary Fig. 2E). All these data indicate that JunD-mediated RPS6KA2 transcription is responsible for BET inhibition resistance.

**JunD/RSK3 signalling correlates to BET inhibition sensitivity.** A previous study implicated that different breast cancer cell lines show contrasting BET inhibition sensitivity[26]. Since the expression of BRD4 shows no significant difference among major breast cancer subtypes[5], we wondered whether the intrinsic level of JunD/RSK3 signalling might be related to BET inhibition sensitivity. To verify this speculation, the expression status of JUND and RPS6KA2 was analysed in four gene expression datasets of breast cancer patients from Gene Expression Omnibus (GEO). Tightly positive correlations of mRNA levels between JUND and RPS6KA2 were observed in different subtypes of breast cancer (Fig. 3a). Interestingly, the positive correlation was also found in patient specimens of ovarian cancer, gastric cancer, pancreatic cancer, prostate cancer and leukaemia (Supplementary Fig. 3A). These data implicate that JunD-dependent transcription of RPS6KA2 is conserved and might act as an universal defensive mechanism upon BET inhibition among multiple cancer types. Secondly, gene differential expression analyses of JUND and RPS6KA2 were conducted based on GEO

datasets. Interestingly, the mRNA levels of the two genes in luminal and HER$^+$ subtype breast cancer were both significantly higher than in BLBC (Fig. 3b). Accordingly, we detected the mRNA and protein expression status of JunD and RSK3 in a panel of breast cancer cell lines. Consistently, JunD and RSK3 both had much higher mRNA and protein levels in luminal and HER2$^+$ breast cancer cell lines (MCF7, BT474 and MDA-MB-453) compared with BLBC cells (SUM1315, MDA-MB-231, BT549 and MDA-MB-157) (Fig. 3c, d). To explore the significance of differential expression of JunD/RSK3, we measured JQ1 sensitivity in the same panel of cell lines following treatment (1 μM) for 48 h. The response to JQ1 varied among these cells, whereby BLBC cells with low JunD/RSK3 levels were highly sensitive and lost almost 50–70% of cell viability, while luminal/HER2$^+$ breast cancer cells with high intrinsic JunD/RSK3 level showed obvious resistance with only a 10–20% decrease of cell viability (Fig. 3e). Lastly, to further confirm the role of JunD/RSK3 signalling in BET resistance, RPS6KA2 and JUND were respectively knocked down in two JQ1-resistant breast cancer cell lines (BT474 and MDA-MB-453) (Supplementary Fig. 3B), and showed that the silencing of both RPS6KA2 and JUND restored the JQ1 sensitivity of these cells (Fig. 3f).

**JQ1 represses BRD4/FOXD3-maintained miR-548d-3p expression.** Previous studies considered that different from other AP-1 family members, the expression of JunD is usually relatively constant even when cells respond to extracellular stress[25]. We were interested in the underlying mechanism of emergent induction of JunD upon BET inhibition. A recent study reported that the processing of a set of primary miRNAs including MIR548D1 is driven by super-enhancer mediated recruitment of Drosha/DGCR8. The BET inhibitor JQ1 preferentially inhibits this process[27]. Interestingly, two mature products of MIR548D1, miR-548d-3p and miR-548d-5p, both have putative binding sites on the 3′UTR region of JUND (Supplementary Fig. 4A). Real-time PCR results revealed the expression of miR-548d-3p in BLBC cells which was markedly downregulated by JQ1, while miR-548d-5p was not detected at all (Fig. 4a). Similar to JQ1, knockdown of BRD4 in MDA-MB-231 and BT549 cells (Supplementary Fig. 4B) obviously repressed the expression of miR-548d-3p (Fig. 4b). Next we asked whether the downregulation of miR-548d-3p contributes to the JQ1-mediated induction of JunD. To test this, we transfected the wild-type 3′UTR luciferase reporter of JUND or its miR-548d-3p-binding-site-mutant into BLBC cells, and found that the addition of miR-548d-3p mimic significantly repressed the luciferase activity of wild-type construct, but not the mutant (Fig. 4c). The addition of miR-548d-3p mimic also repressed JQ1-induced JunD expression, while transfection of the miRNA inhibitor promoted JunD expression

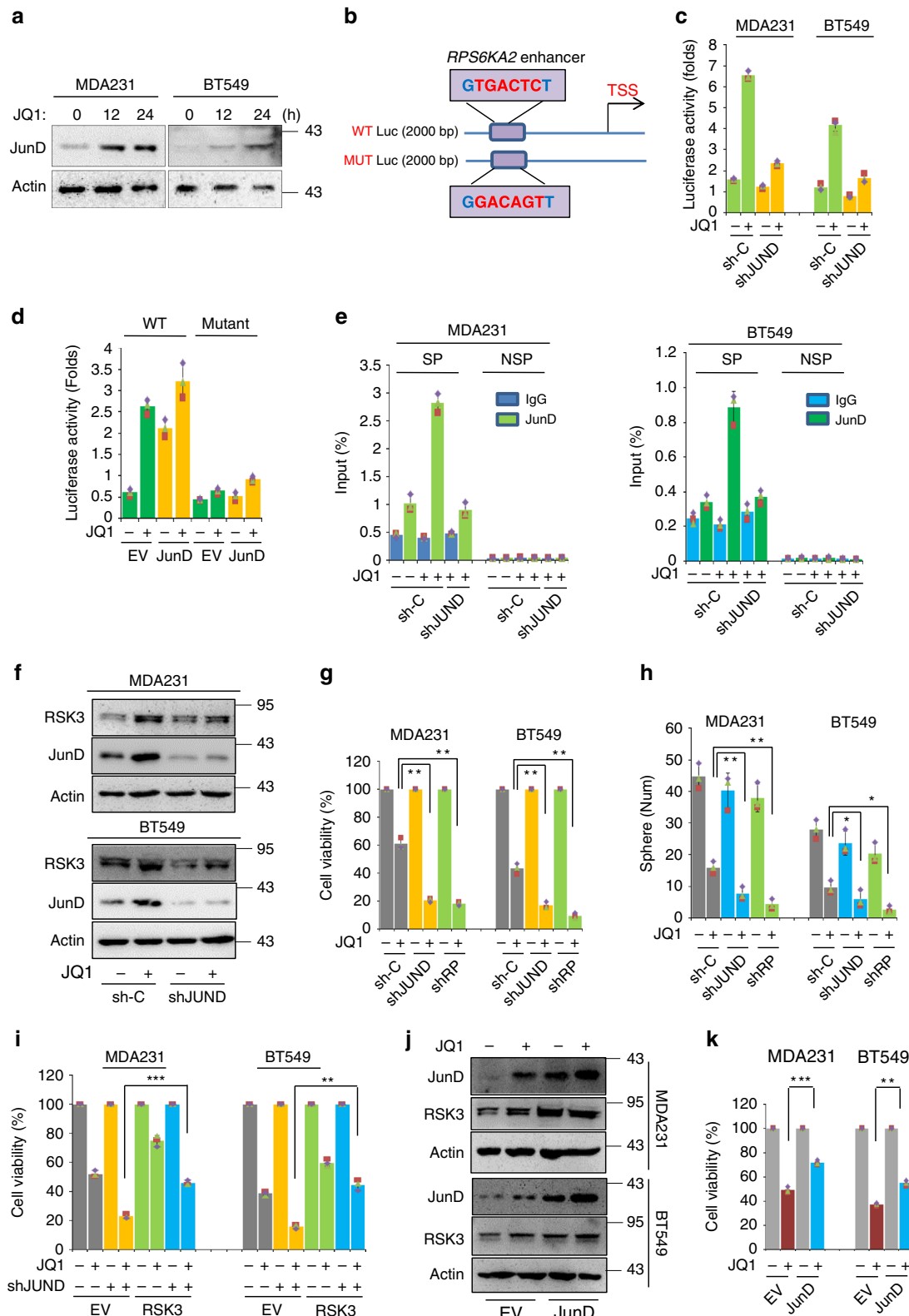

(Fig. 4d and Supplementary Fig. 4C). Similar results were observed regarding RSK3 expression after cells were transfected with miR-548d-3p mimic or inhibitor (Supplementary Fig. 4D). Furthermore, we observed that the knockdown of *BRD4* in MDA-MB-231 and BT549 cells enhanced the JunD and RSK3 expression (Supplementary Fig. 4E). Consistently, the addition of miR-548d-3p mimic enhanced JQ1 sensitivity (Fig. 4e), while the

miRNA inhibitor partially conferred the cells with resistance ability to JQ1 (Fig. 4f). Interestingly, the expression level of miR-548d-3p was highly upregulated in BLBC cells which was confirmed by real-time PCR in an array of breast cancer cell lines (Supplementary Fig. 4F), and bioinformatics analysis of The Cancer Genome Atlas (TCGA) indicated that the expression of *MIR548D1* is uniquely and robustly upregulated in the basal-like

**Fig. 2 JunD-dependent *RPS6KA2* transcription mediates BETi resistance. a** Western blotting was performed to detect JunD protein levels, MDA-MB-231 and BT549 cells were treated with DMSO or JQ1 (1 μM) for 0, 12 and 24 h. **b** Photograph depicted the potential JunD binding site in the enhancer region of *RPS6KA2* gene. Wild-type and mutant *RPS6KA2* gene enhancer luciferase plasmids are shown. **c** Luciferase assays were performed in MDA-MB-231 and BT549 cells transfected with shRNA for JunD and control, in the presence of DMSO or JQ1 (1 μM) for 6 h. Data are reported as mean ± SD. **d** Luciferase assays were performed in HEK293T cells transfected with wild-type or mutant *RPS6KA2* enhancer luciferase construct with or without *JUND* co-transfection. HEK293T cells were treated with DMSO or JQ1 (1 μM) for 6 h. Data are reported as mean ± SD. **e** Chromatin immunoprecipitation (ChIP)-qPCR assay was executed in *JUND*-knockdown BLBC cells and their vector controls treated with DMSO or JQ1 (1 μM) for 6 h. 'SP' indicates specific primers of ChIP for *RPS6KA2* gene enhancer and 'NSP' indicates non-specific primers that recognize the region downstream of the 3′ end of the gene. **f** Western blotting was performed to detect the expression levels of JunD and RSK3 in control and *JUND*-knockdown clones of MDA-MB-231 and BT549. **g** *JUND* or RPS6KA2-knockdown MDA-MB-231 and BT549 cells as well as their vector control cells were treated with DMSO or JQ1 (1 μM) for 48 h, and luminescent cell viability assays were done to detect the killing effects (**$P < 0.01$, one-way ANOVA). **h** Measured tumoursphere formation in *JUND* or RPS6KA2-knockdown MDA-MB-231 and BT549 cells as well as their vector controls. The cells were also treated with DMSO or JQ1 (1 μM). Statistical data of numbers of tumoursphere were shown (*$P < 0.05$, **$P < 0.01$; one-way ANOVA). **i** Rescued expression of RSK3 in control or *JUND*-knockdown BLBC cells which were treated with DMSO or JQ1 (1 μM) for 48 h. Cell viability was measured by CellTiter-Glo® luminescent viability assay (**$P < 0.01$, ***$P < 0.001$; one-way ANOVA). **j** Western blotting was done to examine the expression levels of JunD and RSK3 in control or *JUND*-overexpressing clones of MDA-MB-231 and BT549. **k** *JUND*-overexpressing MDA-MB-231 and BT549 cells as well as their vector control cells were treated with JQ1 (1 μM) for 48 h, cell viability was detected by CellTiter-Glo® luminescent viability assay. Statistical data (mean ± SD) are shown (**$P < 0.01$, ***$P < 0.001$; one-way ANOVA). Source data are provided as a Source Data file.

subtype compared with the other four breast cancer subtypes (Supplementary Fig. 4G), at least partially, explaining why JunD/RSK3 signalling is downregulated in BLBC cells.

BRD4 normally recognizes the acetylated histones and localizes at the promoter or enhancer region of active genes. For example, BRD4 is enriched at the promoter region of G1 phase related genes and maintains their expression[8,9]. In some cases, BRD4 is recruited by transcription factors targeting the promoter region of specific genes and promoting their transcriptional elongation[5,28]. We observed that the mRNA and protein expression of BRD4 has no significant difference among breast cancer subtypes[5], however, the high expression of miR-548d-3p in the BLBC subtype suggests that some other factors, besides BRD4, might determine its particular expression status. Referring to several transcription regulation databases, Forkhead box D3 protein (FOXD3) was predicted to act as a putative transcription factor for *MIR548D1* gene. Some studies have suggested that deficiency of FOXD3 promotes breast cancer progression[29,30], and FOXD3 is also indicated to function as a tumour suppressor in several other cancer types[31–33]. Interestingly, *FOXD3* is also highly expressed in BLBC based on the analysis of TCGA data, like *MIR548D1* (Supplementary Fig. 4H). To study whether FOXD3 and BRD4 are involved in the transcription of the *MIR548D1* gene, a luciferase reporter containing its promoter region (1000 bp before the transcription starting site) was generated, and the potential FOXD3 binding motif (AATTGTTTTTAT) was deleted (Supplementary Fig. 4I). Ectopic expression of FOXD3 or BRD4 stimulated the luciferase reporter activity in HEK293 cells, which was inhibited by JQ1. Deletion of the potential FOXD3 binding motif robustly repressed the FOXD3- and BRD4-induced effects (Fig. 4g). JQ1 treatment and silencing of *BRD4* or *FOXD3* all significantly decreased the luciferase reporter activity in BLBC cells (Fig. 4h). Co-immunoprecipitation experiments revealed the existence of a FOXD3/BRD4 protein complex, which was disrupted by JQ1 (Fig. 4i). Sequential ChIP results further indicated that both FOXD3 and BRD4 were enriched at the same region near to TSS, while JQ1 treatment or *FOXD3* silencing completely expelled BRD4 from the promoter (Fig. 4j). ChIP result also revealed that BRD2 and BRD3 could not associate with the miRNA promoter (Supplementary Fig. 4J). Consistently, knockdown of *FOXD3* (Supplementary Fig. 4K) decreased the expression of miR-548d-3p (Fig. 4k) and induced *JUND* and *RPS6KA2* mRNA expression (Supplementary Fig. 4L), as well as partially reversing the JQ1-mediated killing effect (Fig. 4l). Co-silencing of *JUND* or *RPS6KA2* in *FOXD3*-knockdown clones

restored the JQ1 sensitivity (Supplementary Fig. 4M). To further confirm the exact role of FOXD3 in drug resistance, we overexpressed FOXD3 in luminal breast cancer cell lines (Supplementary Fig. 4N). Ectopic expression of FOXD3 stimulated the expression of miR-548d-3p (Supplementary Fig. 4O) and repressed *JUND* and *RPS6KA2* mRNA expression (Supplementary Fig. 4P), as well as greatly enhancing the BET sensitivity of luminal breast cancer cells (Supplementary Fig. 4Q). All the above data indicate that FOXD3/BRD4 interaction is disrupted by JQ1, leading to reduced miR-548d-3p expression, restoration of JunD, transcription of *RPS6KA2* and BETi resistance. They also reveal that the FOXD3/BRD4/miR-548d-3p axis is highly activated in BLBC cells, which explains the lower JunD/RSK3 levels and higher sensitivity to BET inhibition of BLBC cells compared with luminal and HER2[+] breast cancer cells.

**Targeting EGFR/MEK/ERK reverses BET inhibition resistance.** Next, we attempted to explore the possibility of targeted inhibition of JunD/RSK3 signalling. Pan-RSK inhibitors LJI308 and LJH685 were shown to moderately enhance the JQ1-mediated killing effect (Supplementary Fig. 5A), as they antagonize three RSK proteins and preferentially inhibit RSK1/RSK2. We also detected the synergistic effect of JNKs inhibitor JNK-IN-8, because JunD is a substrate of JNKs and mediates the pro-survival role of the JNK signalling pathway[34]. However, the synergistic effect of JNK-IN-8 was found to be weaker than that of *RPS6KA2* silencing (Supplementary Fig. 5A), probably owing to other kinases that activate JunD, for example, ERK2[35]. We also found that p38 inhibitor SB203580 has no significant capacity to enhance the killing effect of JQ1 (Supplementary Fig. 5B).

Previous studies have demonstrated that phosphorylation is required for the fully activation of newly synthesized RSK3 protein, specifically, its kinase activity and the ability to activate downstream targets is determined by sequential serine/threonine phosphorylation by MAP kinases, including ERK1/2 and ERK5[22–24]. Then we asked whether the inhibition of ERKs might facilitate to overcome BET resistance. Although individual usage of either ERK1/2 inhibitor GDC-0994 or ERK5 inhibitor XMD8-92 still exhibited a weaker synergistic effect than that of *RPS6KA2*-knockdown, the combination was able to fully mimic the knockdown effect (Supplementary Fig. 5C), implicating that co-targeting upstream ERK1/2/5 is an effective way to block the activity of RSK3 and reverse BET resistance.

ERK1/2 and ERK5 mediate the signalling of their upstream molecules, MEK1/2 and MEK5, respectively, which are both

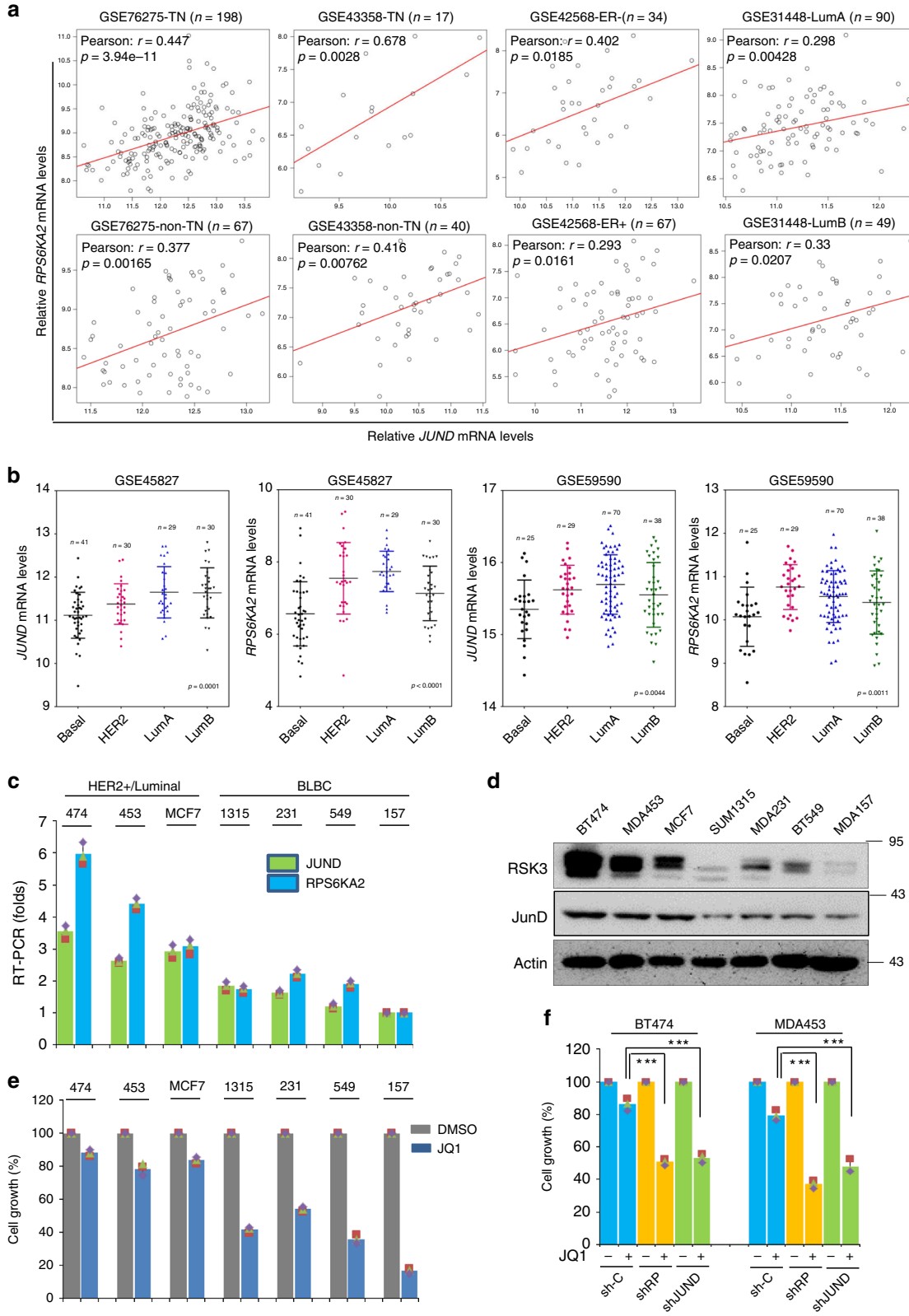

activated by epidermal growth factor receptor (EGFR)[36]. Recently, computational screening was conducted to identify novel dual kinase/bromodomain inhibitors from commercially available small molecules and suggested that EGFR/BET dual inhibition is potentially effective for cancer treatment[37]. Therefore, we wondered whether EGFR inhibition can mimic the effect of combined use of ERK1/2/5 inhibitors to block the

activity of RSK3 and confer BLBC cells with BET sensitivity. To test this, we treated BLBC cells with JQ1 (1 μM) for 48 h in the absence or presence of seven commercially available EGFR inhibitors (1 μM), including osimertinib, AZ5104, erlotinib, icotinib, gefitinib, lapatinib and EAI045. As expected, JQ1 suppressed the viability of BLBC cells by ~50%; although all the EGFR inhibitors did not affect cell viability individually,

**Fig. 3 JunD/RSK3 signalling correlates to BET inhibition sensitivity. a** Correlation analyses of *JUND* and *RPS6KA2* in four datasets of breast cancer patients from GEO. GSE76275 was separated into two sub-groups: triple-negative (TN) and non-triple-negative (non-TN); GSE43358 was separated into two sub-groups: triple-negative (TN) and non-triple-negative (non-TN); GSE42568 was separated into two sub-groups: ER-positive (ER+) and ER-negative (ER−); GSE31448 was separated into two sub-groups: luminal A (lumA) and luminal B (lumB). *Pearson* Coefficients of correlation and *p* values are shown. **b** Gene differential expression analyses of the mRNA levels of *JUND* and *RPS6KA2* were conducted in two datasets that contained four subtypes of breast cancer patients, including BLBC, luminal A, luminal B and HER2+. **c** The mRNA expression statuses of *JUND* and *RPS6KA2* were detected in a panel of breast cancer cell lines by quantitative PCR assay. **d** Protein expression levels of JunD and RSK3 were detected in a panel of breast cancer cell lines by western blotting. **e** Luminal and HER2+ breast cancer cell lines as well as BLBC cell lines were treated with DMSO or JQ1 (1 μM) for 48 h, cell growth was detected by CCK8 assay. Statistical data (mean ± SD) are shown. **f** *RPS6KA2* and *JUND* genes were knocked down in two JQ1-resistant breast cancer cell lines, cell growth was measured by CCK8 assay. Statistical data (mean ± SD) are shown (***P < 0.001, one-way ANOVA). Source data are provided as a Source Data file.

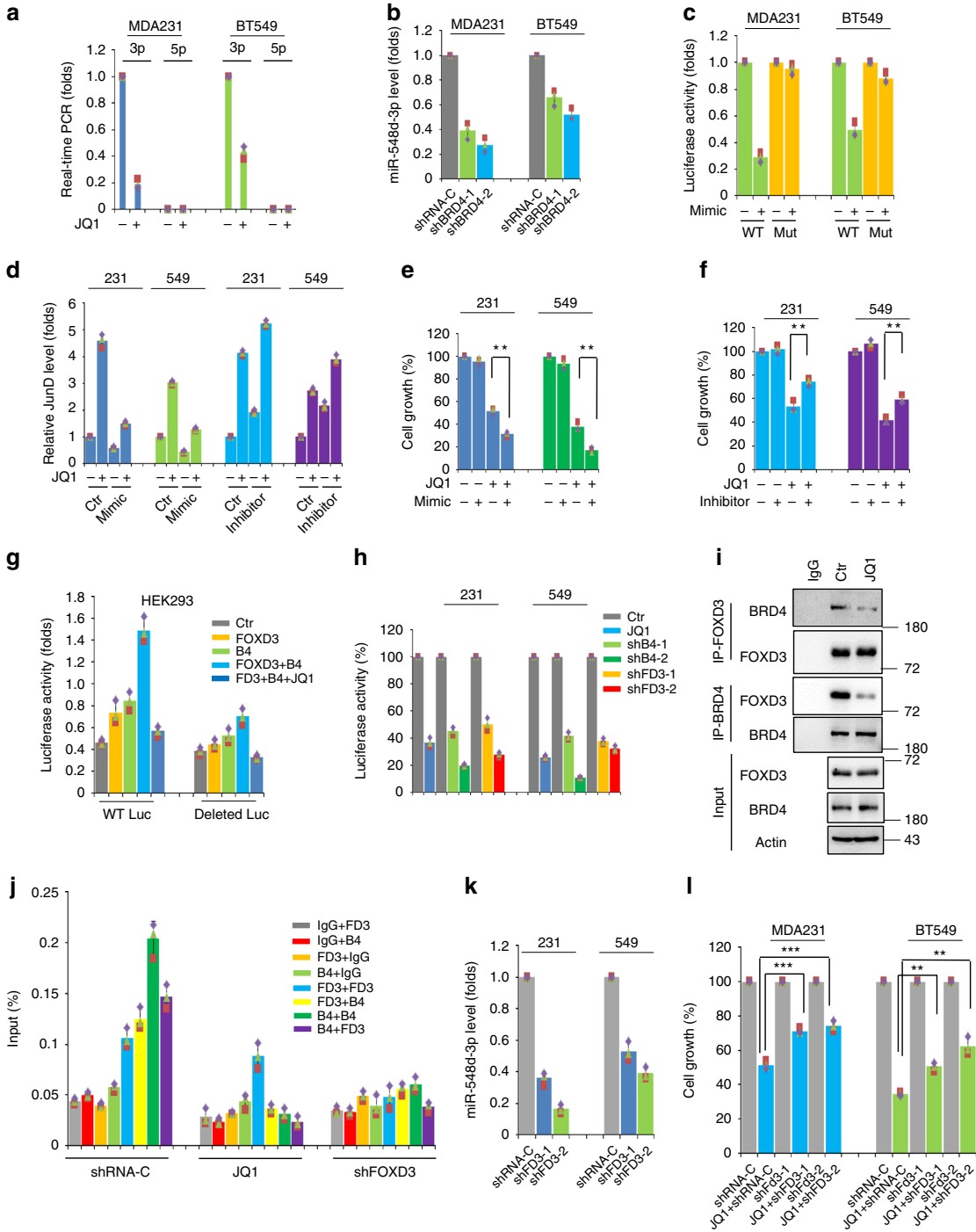

**Fig. 4 JQ1 represses BRD4/FOXD3-maintained miR-548d-3p expression. a** Real-time PCR assay was conducted to detect the expression of miR-548d-3p and miR-548d-5p with or without JQ1 (1 μM) treatment in BLBC cells. **b** Real-time PCR assay was performed to measure the expression of miR-548d-3p in shRNA control and *BRD4*-knockdown clones of MDA-MB-231 and BT549 cells. **c** Wild-type and miR-548d-3p-recognition-mutant *JUND* 3′UTR luciferase reporters were transfected into MDA-MB-231 and BT549 cells with or without miR-548d-3p mimic. Data are reported as mean ± SD. **d** *JUND* mRNA expression was examined in BLBC cells by real-time PCR assay in the absence or presence of mimic or inhibitor of miR-548d-3p. **e** Cell growth was detected with or without miR-548d-3p mimic in BLBC cell lines by CCK8 assay. Statistical data (mean ± SD) are shown (**$P < 0.01$, one-way ANOVA). **f** CCK8 assay was done to detect the effect of miR-548d-3p inhibitor on JQ1 in BLBC cell lines. Statistical data (mean ± SD) were shown (**$P < 0.01$, one-way ANOVA). **g** Luciferase assay was performed in HEK293T cells transfected with WT or deleted *MIR548D1* promoter luciferase construct and *FOXD3* or *BRD4* plasmid. Data are showed as mean ± SD. **h** Luciferase assay was examined in *BRD4*-knockdown, *FOXD3*-knockdown or JQ1-treated MDA-MB-231 and BT549 cells. Data are reported as mean ± SD. **i** Endogenous FOXD3 or BRD4 was pulled down by specific antibodies in MDA-MB-231 cells to observe the FOXD3/BRD4 interaction as detected by immunoprecipitation-western blots. **j** Sequential chromatin immunoprecipitation-QPCR assays were done to measure the enrichment of BRD4 and FOXD3 at the *MIR548D1* gene promoter in control, JQ1 (1 μM) treated, or *FOXD3*-knockdown BLBC cells. **k** Real-time PCR assays were done to examine the expression of miR-548d-3p in vector control and *FOXD3*-knockdown MDA-MB-231 and BT549 cells. **l** CCK8 assays were done in vector control or *FOXD3*-knockdown BLBC cells. Data are reported as mean ± SD (**$P < 0.01$, ***$P < 0.001$; one-way ANOVA). Source data are provided as a Source Data file.

they exhibited markedly synergistic effects with JQ1 to kill the BLBC cells, similar to *RPS6KA2*-knockdown (Fig. 5a).

To determine whether RSK3 is the pharmacological target of EGFR/MEKs/ERKs inhibition, MDA-MB-231 cells were treated with JQ1 (1 μM) and/or osimertinib, GDC-0994, XMD8-92 (1 μM) and GDC-0994/XMD8-92 combination to observe the status of signalling molecules. JQ1 obviously induced JunD and RSK3 protein expression, followed by the elevation of RSK phosphorylation at threonine 359/serine 363 and serine 380; co-treatment of with osimertinib did not compromise the expression of JunD and RSK3, but significantly inhibited the phosphorylation of MEK1/2, ERK1/2, ERK5 and RSK3. GDC-0994 and XMD8-92 blocked the phosphorylation of ERK1/2 and ERK5, respectively, and partially inhibited RSK3 phosphorylation. The combination of GDC-0994 and XMD8-92 also totally abolished RSK3 phosphorylation (Fig. 5b). Osimertinib completely inhibited EGFR phosphorylation, while JQ1 had no significant effect (Supplementary Fig. 5D). These results indicate that ERK1/2/5 phosphorylate and activate JQ1-induced RSK3 protein; EGFR inhibition is able to completely block the activation of RSK3 by simultaneously inhibiting the activities of ERK1/2/5. Consistent with the above observations, osimertinib strongly enhanced JQ1-induced suppression of tumoursphere formation similar to GDC-0994/XMD8-92 (Supplementary Fig. 5E).

Analogously, the combination of MEK1/2 inhibitor U0126 (1 μM) and MEK5 inhibitor BIX-02188 (1 μM) was able to produce similar synergistic effects to JQ1 as osimertinib, but not individually. Furthermore, the JQ1-synergistic effect produced by *RPS6KA2* silencing was phenocopied by the addition of osimertinib, GDC-0994/XMD8-92 or U0126/BIX-02188. It is worth noting that, when *RPS6KA2* had been knocked down in BLBC cells, the extra addition of osimertinib, GDC-0994/XMD8-92 or U0126/BIX-02188 forfeited the synergistic effect with JQ1, indicating their collaborative effects on JQ1 depends on inhibition of RSK3 (Fig. 5c). Consistently, overexpression of RSK3 or JunD partially compromised the killing effects of JQ1/GDC-0994, JQ1/XMD8-92, JQ1/U0126 and JQ1/BIX-02188 ($P < 0.05$). However, the overexpression of both genes were completely unable to counteract the toxic effects of JQ1/osimertinib, JQ1/GDC-0994/XMD8-92 and JQ1/U0126/BIX-02188 ($P > 0.05$), suggesting that the phosphorylation maintained by both ERK1/2 and ERK5 is pivotal for RSK3-mediated BET resistance (Fig. 5d). All the above data indicate that when the expression of RSK3 is induced upon BET inhibition, both the MEK1/2–ERK1/2 and MEK5–ERK5 pathways are required to maintain the activity of RSK3. Although these combined therapies do not compromise the protein expression of RSK3, they shut down the pro-survival role of RSK3 by blocking its activity. Our data also showed that blockade

of either MEK1/2–ERK1/2 or MEK5–ERK5 only partially inhibited the phosphorylation of RSK3; EGFR inhibition completely reversed the RSK3-mediated BET resistance phenotype by dual MEK1/2–ERK1/2 and MEK5–ERK5 inhibition. Interestingly, a previous study reported that the downregulation of protein phosphatase 2A, one of the phosphatases that dephosphorylates RSKs[38], in BET inhibition resistant cells may also contribute to the elevation of phosphorylation level of RSK3[15].

**EGFR inhibition overcomes BET inhibition resistance**. We proceeded to assess the efficacy and applicability of EGFR inhibition in overcoming BETi resistance. Consistently, *BRD4*-knockdown BLBC cells were more sensitive to EGFR inhibition (1 μM) than their vector control cells (Fig. 6a). EGFR inhibition also greatly instigated the anti-tumour effect of other BET inhibitors (iBET151 and iBET762) (Fig. 6b). These two chemicals also strongly induced JunD and RPS6KA2 expression (Supplementary Fig. 6A). Interestingly, EGFR inhibition did not improve the anti-cancer effect of paclitaxel, a major chemo-therapeutic drug for the clinical treatment of BLBC (Fig. 6c), suggesting that EGFR inhibition only targets specific feedback signalling. This combined therapy was also effective in luminal and HER2$^+$ subtype breast cancer cells (Fig. 6d). The sensitivity index was tightly correlated with the endogenous level of RSK3 and JunD (Fig. 3d). To further explore the applicability of combined therapy, we treated OVCAR4 ovarian cancer cells, MGC-803 gastric cancer cells, Panc-1 pancreatic cancer cells and K562 chronic myelogenous leukaemia cells with JQ1 for 48 h in the absence or presence of osimertinib. Similar to BLBC cells, these four cell lines were refractory to EGFR inhibition and obviously responded to individual JQ1 treatment, while EGFR inhibition evoked a strong synergistic killing effect with JQ1. Especially, the combined treatment caused more than 75% inhibition of cell viability in MGC-803 (Fig. 6e). These data suggest the combined therapy may be applicable to treat other cancers that respond to BET inhibition initially but develop drug resistance.

Our previous study indicated that BET inhibitors suppress tumoursphere formation and invasion in BLBC cells[5]. Consistently, we observed JQ1 reduced the number and size of tumoursphere; osimertinib itself did not affect sphere formation but greatly enhanced the JQ1-mediated effects (Fig. 6f and Supplementary Fig. 6B). Similar results were noted in Transwell invasion assay (Supplementary Fig. 6C). Flow cytometry analysis further showed that osimertinib co-treatment robustly exaggerated the cell apoptosis in JQ1-treated BLBC cells (Fig. 6g and Supplementary Fig. 6D). We also observed much stronger caspase-3 cleavage and activation upon the treatment with

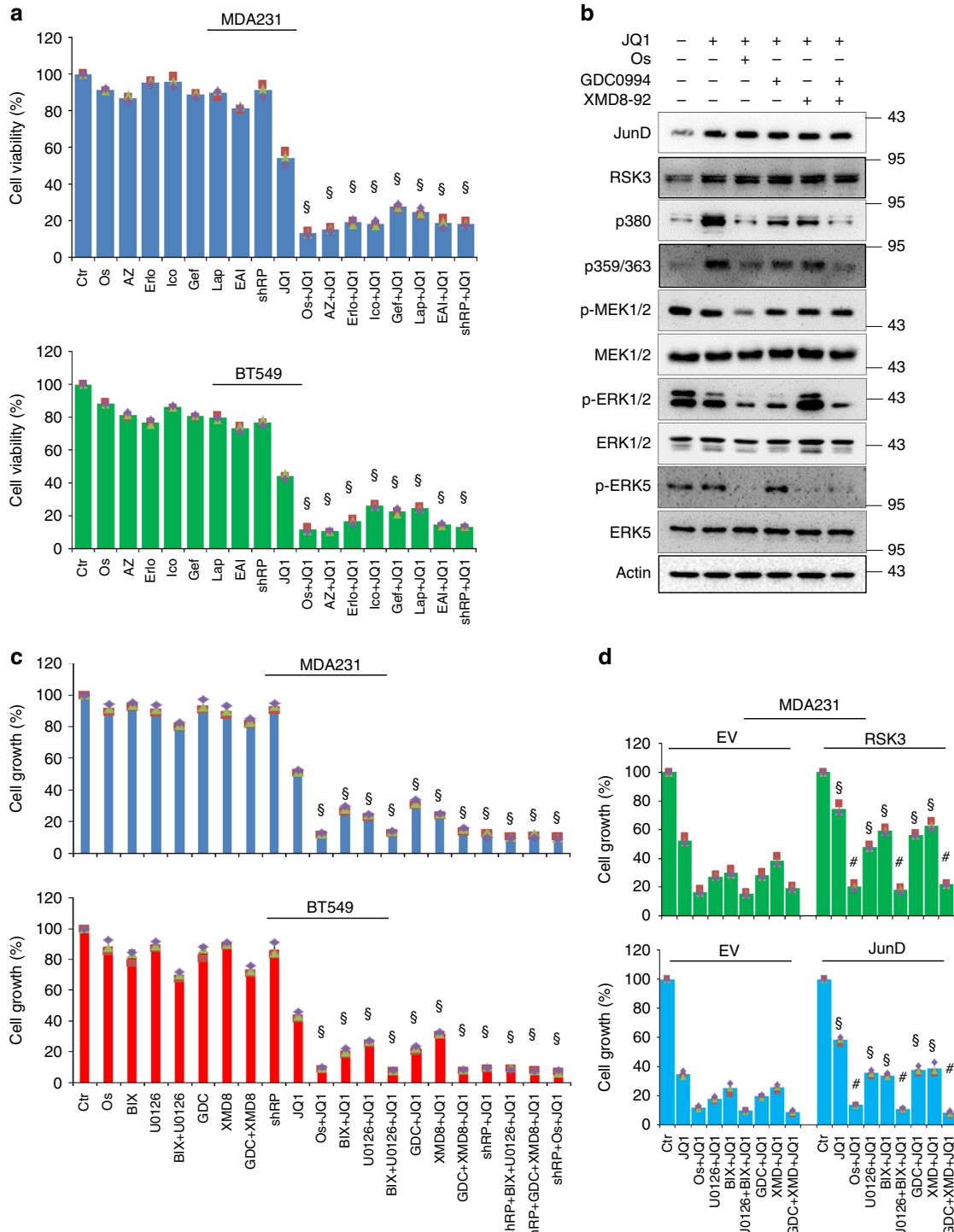

osimertinib/JQ1 compared with either individual treatment (Fig. 6h).

To inspect the potential of the combined therapy in vivo, we implanted MDA-MB-231 cells in nude mice. When xenograft tumours grew to ~100 mm³, mice were equally divided into four groups and administrated with vehicle control, osimertinib (10 mg/kg), JQ1 (35 mg/kg), osimertinib plus JQ1. Single osimertinib-treated tumours grew similarly to vehicle-treated tumours; the size of tumours in the JQ1-treated group was nearly half that of the vehicle-treated tumours. Osimertinib plus JQ1 significantly reduced the size and weight of tumours compared with JQ1 (Fig. 6i–k). Similar results were observed

from MDA-MB-468 derived CCK8 growth assay (Supplementary Fig. 6E) and xenograft mice model (Supplementary Fig. 6F–H). Together, these findings support the concept that EGFR inhibition overcomes BET inhibition resistance.

**JQ1-resistant BLBC cells are sensitive to combined therapies.** To further evaluate the efficacy of combination therapies, we established JQ1-resistant cell clones by stepwise increased concentration of JQ1 in MDA-MB-231 and BT549 cells. Cells that proliferated well at the maximum concentration of JQ1 (20 μM) were ready for analysis[15]. We transfected *RPS6KA2* enhancer luciferase plasmid and empty vector into parental MDA-MB-231

**Fig. 5 Targeting EGFR/MEKs/ERKs reverses BETi resistance. a** Measurement of cell viability of MDA-MB231 and BT549 cells by CellTiter-Glo®
luminescent viability assay. Control and *RPS6KA2*-knockdown cells were treated with JQ1 in the absence or presence of seven EGFR inhibitors for 48 h. 1:
Ctr; 2: osimertinib; 3: AZ5104; 4: erlotinib; 5: icotinib; 6: gefitinib; 7: lapatinib; 8: EAI045; 9: *RPS6KA2*-shRNA; 10: JQ1; 11: JQ1 + osimertinib; 12: JQ1 +
AZ5104; 13: JQ1 + erlotinib; 14: JQ1 + icotinib; 15: JQ1 + gefitinib; 16: JQ1 + lapatinib; 17: JQ1 + EAI045; 18: JQ1 + *RPS6KA2*-shRNA. Statistical data (Mean ±
SD) are shown. *P* values were calculated between single JQ1-treated samples and samples treated with JQ1 plus EGFR inhibitors. Symbol 'S' indicates
statistical significance (*P* < 0.05, one-way ANOVA). **b** MDA-MB-231 cells were treated with JQ1 (1 μM) and/or osimertinib, GDC0994, XMD8-92 (1 μM)
or GDC0994/XMD8-92 for 12 h, western blotting was performed to detect JunD protein expression, and phosphorylated and total levels of MEK1/2, ERK1/
2, ERK5 and RSK3. **c** CCK8 assays were done to measure the killing effects. Control and *RPS6KA2*-knockdown MDA-MB-231 and BT549 cells were treated
with indicated the inhibitors for 48 h. 1: Control; 2: osimertinib; 3: BIX-02188; 4: U0126; 5: U0126 + BIX-02188; 6: GDC0994; 7: XMD8-92; 8: GDC0994 +
XMD8-92; 9: *RPS6KA2*-shRNA; 10: JQ1; 11: JQ1 + osimertinib; 12: JQ1 + BIX-02188; 13: JQ1 + U0126; 14: JQ1 + BIX-02188 + U0126; 15: JQ1 + GDC0994;
16: JQ1 + XMD8-92; 17: JQ1 + GDC0994 + XMD8-92; 18: JQ1 + *RPS6KA2*-shRNA; 19: JQ1 + *RPS6KA2*-shRNA + BIX-02188 + U0126; 20: JQ1 + *RPS6KA2*-
shRNA + GDC0994 + XMD8-92; 21: JQ1 + *RPS6KA2*-shRNA + osimertinib. Statistical data (mean±SD) are shown. *P* values were calculated when
compared between single JQ1-treated samples and samples that were treated with JQ1 plus kinase inhibitors. Symbol 'S' indicates statistical significance
(*P* < 0.05, one-way ANOVA). **d** CCK8 assay was performed to observe inhibition effects. Vector control and RSK3 (*upper*) or JunD-overexpressing
(*bottom*) MDA-MB-231 cells were treated with the indicated inhibitors for 48 h. 1: DMSO; 2: JQ1; 3: JQ1 + osimertinib; 4: JQ1 + U0126; 5: JQ1 + BIX-02188;
6: JQ1 + BIX-02188 + U0126; 7: JQ1 + GDC0994; 8: JQ1 + XMD8-92; 9: JQ1 + GDC0994 + XMD8-92. Statistical data (mean±SD) are shown. *P* values
were calculated when compared between JunD or RSK3-overexpression clones and vector controls in the presence of the same inhibitors. Symbol 'S'
indicates statistical significance (*P* < 0.05, one-way ANOVA), and symbol '#' indicates no significance (*P* > 0.05). Source data are provided as a Source
Data file.

and BT549 as well as their related JQ1-resistant clones; the
luciferase reporter activities were apparently elevated in JQ1-
resistant clones compared with their parental cells, implicating
that JunD/RSK3 signalling is highly activated in JQ1-resistant
clones (Fig. 7a). Western blot results confirmed that JQ1-resistant
stable clones had much higher expression of JunD and RSK3, as
well as the phosphorylated forms of RSK3 (Fig. 7b). Consistently,
miR-548d-3p was downregulated in JQ1-resistant clones (Fig. 7c).
Sequential ChIP analysis further showed that BRD4 was excluded
from the *MIR548D1* promoter in the JQ1-resistant clones com-
pared with their parental cells (Fig. 7d). We then treated these
resistant clones with JQ1 (1 μM) in the absence or presence of
osimertinib, GDC-0994, XMD8-92, GDC-0994/XMD8-92,
U0126, BIX-02188, or U0126/BIX-02188. As expected, these
resistant clones were completely refractory to individual JQ1
treatment, the addition of osimertinib re-sensitized them to JQ1,
and co-treatment of GDC-0994/XMD8-92 or U0126/BIX-02188
had similar effects as osimertinib, followed by either alone
(Fig. 7e). To clarify whether these effects were mediated by RSK3,
we silenced *RPS6KA2* expression in MDA-MB-231- and BT549-
resistant clones (Supplementary Fig. S7). These *RPS6KA2*-
knockdown resistant cells became more sensitive to JQ1 than
control cells; osimertinib, GDC-0994/XMD8-92, and U0126/BIX-
02188 lost the synergistic effect with JQ1 in *RPS6KA2*-knock-
down resistant cells, indicating RSK3 is the exact and critical node
for pharmacological intervention (Fig. 7f). These data indicate
that the targeted strategies towards RSK3 are capable of killing the
tumour cells that have developed resistance to BET inhibition.

## Discussion
Our study indicates the molecular mechanism of BET inhibition
resistance and implicates therapeutic strategies for treatment of
BLBC. BET inhibitors have demonstrated significant efficacy in
multiple cancer models, including BLBC. However, the anti-
tumour efficacy of BET inhibitors is still limited; the development
of BET inhibitor resistance in cancer cells is a pressing problem,
and its mechanism, especially at the level of transcriptional
induction of compensatory survival signalling, has not been
explained. Integrating approaches in mRNA profiling, signal
transduction, and molecular and chemical biology, we provide a
mechanism of drug resistance in which BLBC cells struggle to
survive in JQ1 treatment by means of inducing *RPS6KA2*
expression through JunD-mediated responsive transcription. This
is achieved by miR-548d-3p repression due to the disruption of
BRD4 from the FOXD3-localized *MIR548D1* gene promoter. Two

parallel signalling pathways downstream of EGFR, including
MEK1/2–ERK1/2 and MEK5–ERK5, phosphorylate and activate
the inducible RSK3 protein, eventually leading to the enrichment
of activated RSK3 protein in tumour cells to counteract the acute
killing effect of JQ1. Our data further show that individual
inhibition of MEK1/2–ERK1/2 or MEK5–ERK5 was not able to
completely repress the elevated RSK3 activity in JQ1-treated cells;
EGFR inhibition abolished both signalling pathways simulta-
neously and completely shut down the RSK3 activity. In all, we
demonstrate that by combining JQ1 with inhibitors of EGFR or
MEK–ERK, the BET inhibitor resistance phenotype of BLBC cells
is able to be reversed, proposing that these therapeutic combi-
nations are clinically effective in the treatment of patients with
BLBC (Fig. 7g). Interestingly, this reversal in phenotype is specific
for BETi resistance, as chemotherapy resistant cells remain
refractory to EGFR inhibition. The anti-tumour effect of dual
BET/EGFR inhibition is also significant in a series of other types
of cancer, including luminal/HER2[+] breast cancer, gastric cancer,
pancreatic cancer, ovarian cancer and chronic myelogenous leu-
kaemia, reflecting its great potential value in cancer treatment.

As a leucine zipper DNA-binding protein, JunD is a primary
member of the Jun family that are crucial components of AP-1
transcription factor[25], which is involved in basal-like pre-
malignancies[39]. However, its role in drug resistance is barely
known. Here we identified its direct target gene, *RPS6KA2*, under
BET inhibition based on biochemical and molecular evidence. The
expression of *JUND* and *RPS6KA2* is tightly and positively cor-
related in various cancer types. The overexpression or knockdown
of *JUND* and following rescue expression of *RPS6KA2* further
demonstrated that the prompt induction of JunD participated in
the expression of RSK3, drug resistance and the survival of BLBC
cells under BET inhibition. Previous studies have indicated that
although Jun proteins (c-Jun, JunB, and JunD) share similar
DNA-binding affinity, their expression patterns vary greatly in
response to stress. Normally, JunB and c-Jun function as
immediate-early response genes that are robustly induced by
extracellular stimulus. However, unlike the other two Jun proteins,
the expression of JunD usually remains relatively constant even
under stress[25]. Intriguingly, we observed the expression of JunD
was induced promptly upon JQ1 treatment, which was a result of
the repression of miR-548d-3p by JQ1. *MIR548D1* is a non-coding
RNA gene that produces two mature microRNAs, miR-548d-5p
and miR-548d-3p, both of which are putative microRNAs that
target JunD. This was confirmed by results showing that miR-
548d-3p targets the 3′UTR region of *JUND* and represses its

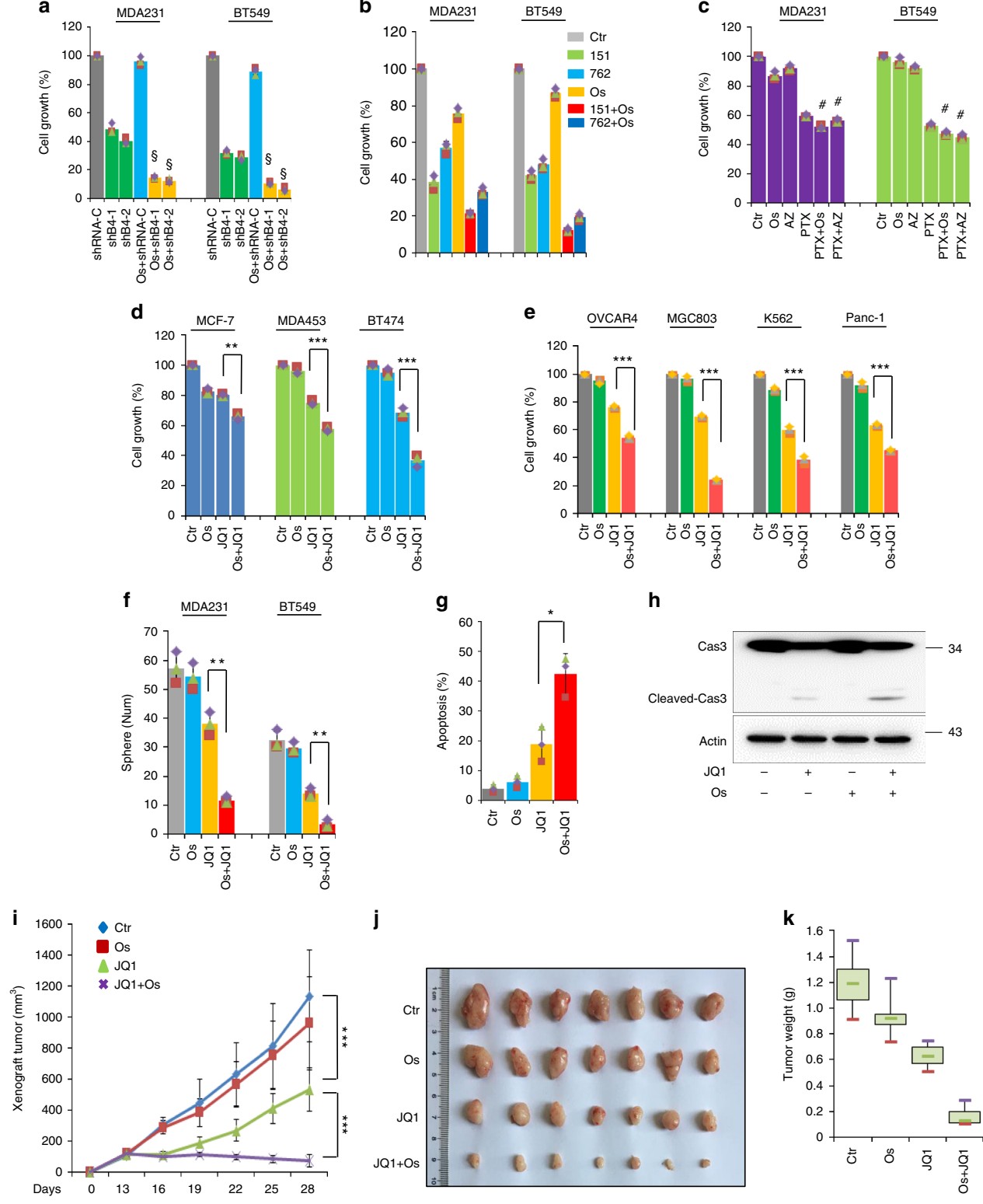

expression. JQ1 treatment and *BRD4* or FOXD3 silencing all significantly repressed the expression of miR-548d-3p. Next, luciferase and sequential ChIP assays indicated that BRD4/FOXD3 was enriched in the promoter region of *MIR548D1*. JQ1 treatment completely displaced BRD4 from the FOXD3-localized promoter region, which lead to the loss of ability of BRD4/FOXD3 to maintain the expression of miR-548d-3p and the consequent restoration of JunD; this conclusion was also supported by the evidence that knockdown of *BRD4* or *FOXD3* upregulates the

expression of JunD. These data indicate that BET inhibitors not only disrupt the interaction of BRD4 with pro-oncogenic transcription factors and suppress oncogene expression[5,28], but also ameliorate the BRD4/FOXD3 complex maintained miR-548d-3p expression, eventually leading to BET inhibition resistance. Therefore, we propose that the pathological role of BRD4 is dependent on its binding partners and cellular context.

Our study identifies *RPS6KA2* as an essential BET inhibition resistance gene whose expression and activity are modulated by

**Fig. 6 EGFR inhibition overcomes BETi resistance. a** Vector control or *BRD4*-knockdown cells were treated with DMSO or osimertinib for 48 h, and the effects on proliferation were detected by CCK8 assay. Statistical data (mean ± SD) are shown. *P* values were calculated when compared between *BRD4*-knockdown clones treated with vehicle and osimertinib. Symbol '§' indicates statistical significance ($P < 0.05$, one-way ANOVA). **b** CCK8 assay was used to detect the viability of MDA-MB231 and BT549 cells when the cells were treated with BET inhibitors (iBET151 and iBET762) and/or osimertinib (**$P < 0.01$, ***$P < 0.001$; one-way ANOVA). **c** MDA-MB231 and BT549 cell proliferation was observed by CCK-8 assay when the cells were treated with paclitaxel as well as osimertinib or AZ5104. Statistical data (mean ± SD) are shown. *P* values were calculated when compared between paclitaxel alone, paclitaxel/osimertinib or paclitaxel/AZ5104. Symbol '#' indicates no statistical significance ($P > 0.05$, one-way ANOVA). **d** Luminal and HER2$^+$ breast cancer cell lines were treated with JQ1 (1 μM) for 48 h in the absence or presence of osimertinib (1 μM), the effects were detected by CCK8 assay (**$P < 0.01$, ***$P < 0.001$; one-way ANOVA). **e** Ovarian cancer cell line OVCAR4, gastric cancer cell line MGC-803, pancreatic cancer cell line PANC-1 and chronic myelogenous leukaemia cell line K562 were treated with JQ1 and/or osimertinib (1 μM) for 48 h, and cell growth was measured by CCK8 assay (***$P < 0.001$, one-way ANOVA). **f** Tumoursphere formation in MDA-MB-231 and BT549 cells upon treatment with osimertinib and/or JQ1 was detected. Statistical data are shown (**$P < 0.01$, one-way ANOVA). **g** FITC/annexin V staining-based flow cytometry analysis was done in JQ1 and/or osimertinib-treated MDA-MB-231 cells (*$P < 0.05$, one-way ANOVA). **h** Detection of cleaved and total caspase-3 upon treatment with osimertinib and/or JQ1 in MDA-MB-231 cells. **i–k** MDA-MB-231-derived xenograft mice were separated into four groups ($n = 7$), and administered with vehicle control, osimertinib (10 mg/kg), JQ1 (35 mg/kg), osimertinib plus JQ1. **i** Growth curves of xenograft tumour are shown (***$P < 0.001$); **j** Photographs of tumours; **k** Tumour weight. Source data are provided as a Source Data file.

an intrinsic signalling network, and as a critical node for pharmacological intervention. RSK3 has been implicated to be involved in the regulation of diverse cellular processes, including cell-cycle progression[22]. However, whether *RPS6KA2* is an inducible drug resistance gene and how it is transcriptionally modulated remain unclear. We identified that JunD governs the early-stage transcriptional induction of *RPS6KA2* in response to JQ1 treatment based on multiple evidence. All these data plus the results from experiments of gene overexpression or silencing, demonstrate that RSK3 mediates resistance to BET inhibitors in BLBC cells. The addition of EGFR, MEK or ERK inhibitors, which totally or partially blocked the phosphorylation of RSK3, exaggerated the responsiveness of tumour cells to JQ1. These observations are consistent with some recent studies reporting that combination of MEK1/2 and BET inhibitors favours the treatment of several types of cancer[40–43]. However, our findings further demonstrate that BLBC cells survive under BET inhibition through the activation of inducible RSK3 by parallel pathways, MEK1/2–ERK1/2 and MEK5–ERK5 signalling. Our data also suggest that directly targeted RSK3 inhibition is more effective, because only combined inhibition of ERK1/2/5 or MEK1/2/5 results in a similar degree of decreased proliferation and augmented apoptosis compared with the effect of EGFR inhibition or *RPS6KA2* silencing. Therefore, the robust efficacy observed in our models provides the rationale for the development of a specific RSK3 inhibitor in combination with BET inhibitors or a dual BRD4–RSK3 inhibitor for BLBC treatment. Interestingly, BLBC cells usually have lower endogenous level of JunD/RSK3 signalling and are more sensitive to JQ1 than other subtypes of breast cancer; silencing of *JUND*/*RPS6KA2* can restore the sensitivity in JQ1-resistant luminal or HER2$^+$ breast cancer cells, suggesting that besides adaption to BET inhibition, JunD/RSK3 signalling is also an intrinsic safeguard mechanism.

Although BLBC cells lack oestrogen receptor, progesterone receptor and human EGFR 2 expression, EGFR protein is frequently overexpressed in BLBC[44]. EGFR is a receptor tyrosine kinase of the ERBB family, which triggers various downstream signalling pathways, including Ras–Raf–MEK–ERK, leading to cell proliferation and survival[45,46]. EGFR is also overexpressed in several of cancer types, including lung, colon, head and neck, brain and pancreatic cancers, and is responsible for their development and progression[47]. Some inhibitors of EGFR, including small tyrosine kinase inhibitors and monoclonal antibodies, are currently applied in the clinic for cancer treatment, such as non-small cell lung cancer[48,49]. However, the perspective of anti-EGFR therapy in BLBC remains obscure because the clinical trials of EGFR inhibitors in BLBC, including monotherapy and in combination with

chemotherapy, have failed due to low response rates[44]. One plausible reason for the lack of response is that under normal conditions, the proliferation and survival of BLBC cells are not exclusively dependent on EGFR signalling. Recently, a computational screen suggested that EGFR inhibition might lead to synergistic lethality with BET inhibitors[37]. Therefore, we speculated that EGFR signalling is possibly involved in tumour cell survival under the pharmacological pressure of BET inhibition. In this study, we assessed the synergistic effects of EGFR inhibitors on BET inhibition and drew the conclusion that the activation of RSK3 upon BET inhibition restores the dependence of survival of BLBC cells on EGFR signalling. EGFR inhibition mimics the dual inhibition of MEK/ERK, effectively targets JunD/RSK3 and reverses the BETi resistance phenotype, again highlighting the potential of anti-EGFR therapies in treating BET-inhibition-resistant tumours.

## Methods

**Cell culture.** Breast cancer cell lines MDA-MB-231, MCF-7, MDA-MB-157, BT474 and MDA-MB-453 were cultured in DMEM medium plus 10% FBS; BT549 cells were grown in RPMI-1640 plus 10% FBS. SUM1315 cells were cultured in Ham's F-12 medium plus 5% FBS with 10 ng/ml EGF, and 10 μg/ml insulin. All breast cancer cells were purchased from ATCC. For selection of stable clones, puromycin (1.0 μg/ml) was used. JQ1-resistant cancer cell clones were obtained by treatment with stepwise increased concentrations of JQ1. MDA-MB-231 and BT549 cells were incubated with 1 μM JQ1 for 2 days, and then the medium was replaced with fresh medium without JQ1 until the cells had recovered. For each sub-culture, the cells were incubated with gradually increasing concentrations of JQ1 for 2 days and cultured without JQ1 until the cells grow well. Cells that grew well at the maximum concentration of JQ1 (20 μM) were deemed as JQ1-resistant clones and stored for further analyses.

**Reagents.** Antibodies against BRD4 (#13440), phospho-EGFR (#3777), EGFR (#4267), phospho-RSK3 (Thr359/Ser363) (#9344), phospho-RSK3 (Ser380) (#11989), phospho-MEK1/2 (#9154), MEK1/2 (#8727), phospho-ERK1/2 (#4370), ERK1/2 (#4695), phospho-ERK5 (#3371), ERK5 (#3552), c-Fos (#2250), c-Jun (#9165) and caspase-3 (#14220) were purchased from Cell Signalling (Danvers, MA). Antibodies against RSK3 (GTX111071) and JunB (GTX79258) were from GeneTex (Irvine, CA). JunD antibody (#710701) was obtained from Thermo Fisher (Waltham, MA). All antibodies were used at 1:1000 dilutions. ShRNA against *BRD4* and *RPS6KA2* were purchased from Sigma-Aldrich (St. Louis, MO). PLKO.1 shJUND-puro (TRCN0000014974) was obtained from Addgene (Cambridge, MA). Inhibitors of BRD4, EGFR, MEK1/2, MEK5, ERK1/2, ERK5, JNKs and RSKs were purchased from Selleckchem (Houston, TX). Lenti-virus plasmids expressing *RPS6KA2* and *JUND* were purchased from Genechem (Shanghai, China) and Genecopoeia (Guangzhou, China), respectively. Reporter plasmids expressing wild-type and mutant *RPS6KA2* gene promoter, and wild-type and mutant 3′UTR of *JUND* were obtained from Genecopoeia (Guangzhou, China). The mimic and inhibitor of miR-548d-3p were purchased from GenePharma (Guangzhou, China).

**Immunoblotting and immunoprecipitation.** Cell lysates were extracted using IB buffer (50 mM Tris-HCl (pH7.4), 150 mM NaCl, 0.2 mM EDTA, 0.2% NP40, 10% Glycerol, protease and phosphatase inhibitors), and immunoprecipitated with the indicated antibodies and Protein G-Sepharose (Thermo). Pulldown protein complexes were analysed by western blot.

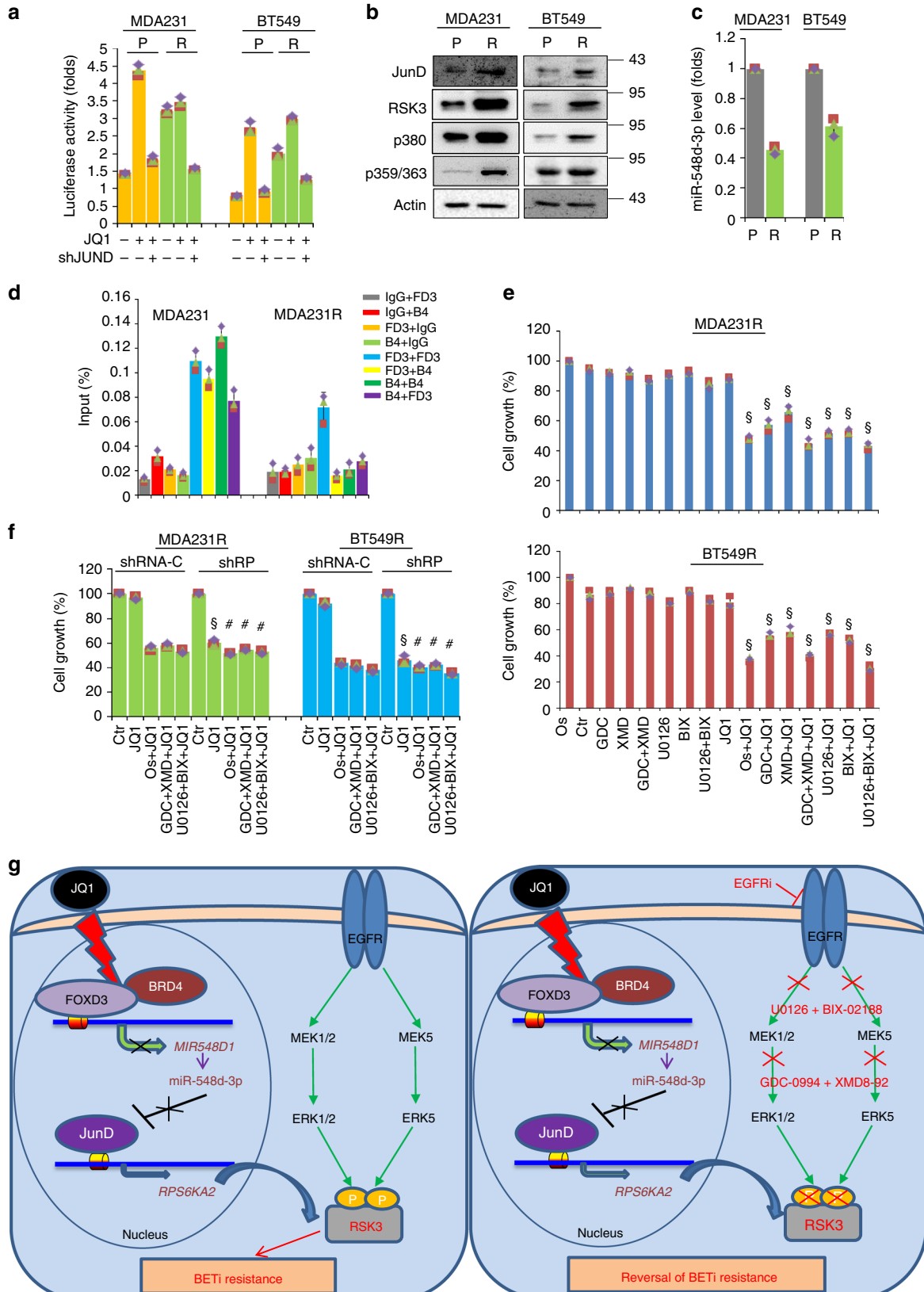

**Real-time PCR**. RNA was extracted from cells by RNeasy Mini Kit (#74104, Qiagen, Valencia, CA), and was reverse transcribed using SuperScript[R] III Reverse Transcriptase (#18080044) from Thermo Fisher Scientific (Waltham, MA). Real-time PCR was analysed using Power SYBR Green Master Mix (Applied Biosystems). For micro-RNA detection, small RNAs were extracted from tumour cells by small RNA isolation reagent (Takara, #9753 A). Bulge-loop miRNA qRT-PCR

PrimerSets (one RT primer and a pair of qPCR primers for each set) specific for miR-548d-3p and miR-548d-5p were designed by RiboBio (Guangzhou, China). Then, the miRNA bulge-loop was reverse transcribed with the First-Strand cDNA Synthesis Kit (Thermo Fisher Scientific) and quantified by qPCR using SYBR Green Real-Time PCR Master Mix Kit (Applied Biosystems) according to the indicated manufacturer's instructions.

**Fig. 7 JQ1-resistant BLBC cells are sensitive to combined therapies. a** *RPS6KA2* gene enhancer luciferase reporter activity was detected in two resistant clones as well as their parental cells. The data are reported as mean ± SD. 'P' indicates parental BLBC cells, and 'R' indicates resistant clones. **b** Measurement of the protein levels of JunD and RSK3 as well as the phosphorylated forms of RSK3 in resistant clones and parental BLBC cells. **c** MiR-548d-3p levels were measured by RT-PCR in resistant clones and parental BLBC cells. **d** Sequential chromatin immunoprecipitation was performed in parental and JQ1-resistant BLBC cells. **e** MDA-MB-231 and BT549 JQ1-resistant clones were treated with the indicated inhibitors for 48 h, and the effects were detected by CCK8 assay. 1: DMSO; 2: osimertinib; 3: GDC0994; 4: XMD8-92; 5: GDC0994 + XMD8-92; 6: U0126; 7: BIX-02188; 8: BIX-02188 + U0126; 9: JQ1; 10: JQ1 + osimertinib; 11: JQ1 + GDC0994; 12: JQ1 + XMD8-92; 13: JQ1 + GDC0994 + XMD8-92; 14: JQ1 + U0126; 15: JQ1 + BIX-02188; 16: JQ1 + BIX-02188 + U0126. Statistical data (mean ± SD) are shown. *P* values were calculated compared between single JQ1-treated samples and samples that treated with JQ1 plus kinase inhibitors. Symbol '§' indicates statistical significance ($P < 0.05$, one-way ANOVA). **f** Cell growth was detected by CCK8 assay in vector control and *RPS6KA2*-knockdown JQ1-resistant clones. Cells were treated as following, 1: DMSO; 2: JQ1; 3: JQ1 + osimertinib; 4: JQ1 + GDC0994 + XMD8-92; 5: JQ1 + BIX-02188 + U0126. Statistical data (mean ± SD) are shown. *P* values were calculated when compared between *RPS6KA2*-knockdown clones and vector controls in the presence of the same inhibitors. Symbol '§' indicates statistical significance ($P < 0.05$, one-way ANOVA), and symbol '#' indicates no significance ($P > 0.05$). **g** A proposed model illustrating the underlying mechanism of BET inhibition resistance and potential combination therapies. Source data are provided as a Source Data file.

**Chromatin immunoprecipitation.** Approximately $1 \times 10^6$ control MDA-MB-231 and BT549 cells as well as treated cells or stable clones were fixed with cross-link solution and collected, ChIP assays were performed using Imprint Chromatin Immunoprecipitation Kit (Sigma, #CHP1) according to the manufacturer's instructions. Antibody-immunoprecipitated DNA was analysed by real-time PCR. Sequential ChIP assay was performed using the Re-ChIP-IT magnetic chromatin reimmunoprecipitation kit (Active Motif, Carlsbad, CA) according to the manufacturer's protocol. Briefly, the chromatin–IgG, chromatin–FOXD3 or chromatin–BRD4 complex was re-immunoprecipitated using anti-BRD4, anti-FOXD3 or anti-IgG antibodies. After the Re-ChIP assay, the isolated DNA was analysed by quantitative RT-PCR.

**Luciferase reporter assay.** Wild-type and mutant *RPS6KA2* enhancer luciferase reporter plasmid were constructed by cloning the 2000 bp fragment before TSS into pEZX-FR01 plasmid. *MIR548D1* promoter luciferase reporter plasmid was constructed by cloning the 1000 bp fragment before TSS into pEZX-FR01 plasmid. The 747 bp wild-type and mutant 3′UTR region of *JUND* gene were cloned into pEZX-MT06 plasmid (Genecopoeia, Guangzhou, China). Cells were seeded in 60 mm dishes and transfected with mentioned plasmids using FuGene 6 transfection reagent (Roche) for 24 h. Cell lysates were extracted and luciferase activity was measured using the Luc-Pair™ Duo-Luciferase Assay Kit 2.0 (Genecopoeia, Guangzhou, China). All experiments were performed three times in triplicate. Relative luciferase activities were calculated as fold induction compared with vector control.

**Cell viability and growth assay.** Cells were seeded at 3000 cells per well in growth media, allowed to adhere overnight, and treated with test compounds for the indicated time. Cell viability and growth potential were determined using CellTiter-Glo kit (Promega, USA) and CCK-8 kit (Selleck Chemicals, USA), respectively, and results were represented as background-subtracted relative light units normalized to a dimethyl sulfoxide (DMSO)–treated control. Statistical analysis (mean ± SD) with triplicates is shown.

**FACS assay for apoptosis detection.** The tested cells were washed twice with PBS followed by re-suspension in binding buffer at a concentration of $1 \times 10^6$ cells/ml. Then 500 μl of the apoptotic cell suspension was placed in a plastic $12 \times 75$ mm test tube and Annexin V-FITC conjugate and propidium iodide was then added to each cell suspension. The tubes were incubated at room temperature for exactly 10 min in the dark. The fluorescence of the cells was immediately determined with a flow cytometer.

**Tumoursphere assay.** $1 \times 10^4$ cells were plated in a single-cell suspension on ultra-low attachment plates (Corning) in DMEM/F-12 medium supplemented with 20 ng/ml EGF, 10 μg/ml insulin, 0.5 μg/ml hydrocortisone and B27. Tumour-spheres were counted and images were taken after 5–7 days. All experiments were performed in triplicate.

**Invasion assay.** $1 \times 10^5$ BLBC cells were seeded in the upper Boyden chamber of Transwell plate coated with Matrigel (BD biosciences, San Jose, CA) while the bottom chamber was filled with non-serum culture medium plus 100 nM LPA. After incubation for 24 or 48 h, the invasive cells were stained with crystal violent and counted.

**Bioinformatics analysis.** Microarray gene expression data for patients with breast carcinoma, ovarian cancer, prostate cancer, gastric carcinoma, pancreatic cancer and leukaemia were downloaded from the GEO database and TCGA. The Pearson's correlation coefficient was used to quantify the correlation. *P* values were calculated based on testing the hypothesis of correlation coefficient equal to zero, i.e., the expression of genes was independent.

**RNA sequencing.** Total RNA was extracted from vehicle and JQ1 (1 μM) treated MDA-MB-231 cells using TRIzol and QIAGEN RNeasy mini kit. RNA sequencing was performed by Novogene (Beijing, China). The genes that encode kinase that were upregulated significantly in JQ1-treated samples were selected.

**Mice xenograft model.** Animal experiments were performed in accordance with the approval of the Southern Medical University animal care and use committee. Female BALB/c nude mice (4–6 weeks) were purchased from Guangdong Medical Laboratory Animal Center. Mice were housed in autoclaved, ventilated cages and provided with autoclaved water. $1 \times 10^6$ MDA-MB-231 vector control cells or corresponding clones were injected subcutaneously ($n = 7$ for each group). JQ1 (35 mg/kg) or osimertinib (10 mg/kg) was administered every 2 days. For MDA-MB-468 cells, $4 \times 10^6$ cells were injected subcutaneously ($n = 7$ for each group). JQ1 (35 mg/kg) or osimertinib (10 mg/kg) was administered every 3 days. Tumour growth was monitored with calliper measurements; tumour volume was calculated according to the formula: $length \times width^2/2$. Mice were euthanized, tumours were weighed and images were taken.

**Statistical analysis.** Data are presented as mean ± SD. A Student's *t* test (two-tailed) was used to compare two groups, which satisfy normal distribution with homogeneous variance. Multiple comparisons were analyzed by one-way ANOVA and Welch's test was used for data with unequal variance. *P* values of <0.05 were considered significant, and \**P* < 0.05, \*\**P* < 0.01 and \*\*\**P* < 0.001.

**Reporting summary.** Further information on research design is available in the Nature Research Reporting Summary linked to this article.

## Data availability
RNA-sequencing data are available at the GEO data repository with the accession code GSE140003. The authors declare that all data supporting the findings of this study are available within the article and its Supplementary Information files and from the corresponding author on reasonable request. The source data underlying Figs. 1b–f, 2c–k, 3c, e–f, 4a–l, 5d, 6a–g and 7a, c, f and Supplementary Figs. 1b, g, 2c, e, 4e–q and 6e, h are provided as a Source Data file.

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

## Acknowledgements

This work was supported by the National Natural Science Foundation of China (81672629 and 81872168), and the Science and Technology Program of Guangzhou, China (201707010331) to J.S.

## Author contributions

J.S. conceived and designed the project, analysed the data and wrote the manuscript. F.T., K.G., K.S. and Y.L.H. performed the experiments and bioinformatics analyses.

## Competing interests

The authors declare no competing interests.
