## [Transparent Peer Review File · Nature Communications]

Reviewers' comments:

Reviewer #1 (Remarks to the Author); expert in BET inhibitors resistance:

The authors investigate the mechanism of JQ1 resistance in the breast cancer. They found that JQ1 treatment evicts BRD4 from the Foxd3 localized miRNA promoter and repress the expression of miR-548d-3p. Furthermore, they showed that miRNA loss causes the restored JunD and RSK3. Dual inhibition of ERK or EGFR inhibition overcomes the BET inhibitor resistance. Findings are very novel, interesting and mechanistic. However, there are a few issues need to be addressed.

Major points:

1. In the paper, the authors defined a mechanism of JQ1 resistance in breast cancer cells. How about other BET inhibitors, like iBET151 and iBET762. The authors should repeat some key experiments using the other iBET inhibitors.
2. The authors found BRD4 can bind to miRNA's promoter. How about BRD2 and BRD3? Do they bind to the same promoter? Does JQ1 or iBET also block the BRD2 and BRD3 binding in this promoter?
3. The authors found that JunD can bind to the promoter of RSK3. JunD is the member of AP-1. AP-1 also has other members, including Fos, c-Jun and JunB, they share the similar binding motif (Hernandez, J.M. et al, Oncogene 2008). The authors should rule out other members' function in the RSK3 promoter.
4. Also for the RSK3 promoter, the -1771 bp is not in the region of promoter, the promoter region is -800 - +200bp (Kwon et al, PNAS, 2007). It should be an enhancer region or the authors should show the H3K4me1 and H3K4me3 data in the breast cancer cells.
5. In Figure 2E, the ChIP assay cannot demonstrate the direct binding for proteins and DNAs. The authors should test the direct binding using EMSA.
6. In Figure 3C and 3D, MCF-7 is not a HER2+ cell line (K Subik et al., 2010).
7. In Figure S5, the authors should rule out the other MAPK members, like JNK and p38.

Minor points:

1. In figures 3A and S3A, some of the labeled words are too small to be seen clearly.

Reviewer #2 (Remarks to the Author); expert in breast cancer signalling:

Tai et al present a nice molecular report in which they delineate a pathway explaining some of the preferential activity of BET inhibitors (specifically JQ1) toward TNBC/BLBC. The authors use two TNBC cell lines primarily to demonstrate this effect: MDA-231 and BT549. They demonstrate very thoroughly a mechanism whereby FoxD3 and BRD4 interactions are disrupted by JQ1 leading to reduced transcription of MiR548D1, leading to loss of JunD regulation. JunD-mediated transcription of RSK3 drives therapeutic resistance, apparently requiring upstream activation of ERK1/2/5, which can be blocked by broad-spectrum MEK inhibition or EGFR inhibition (at least in MDA-231). The data are quite thoroughly done and include both chemical and genetic inhibition, although the latter is inferior (weakly repressive shRNAs complicate the analyses). Although I would not request the authors to repeat the entire work using something more clean (better shRNAs or CRISPR), I would suggest they use better genetic inhibitors in the future. Overall the data presented support a nice complete mechanism, although the therapeutic upshot with more thorough and robust in vivo models is a missed opportunity.

Major points:

- 1) At least a few of the linear mechanisms should be shown with a different BET inhibitor (unless I missed it) to confirm this is a class-specific effect

2) The only tumor model used in the manuscript is MDA-231, which, while widely used as a 'model' BLBC cell line, most breast cancer researchers agree that they are somewhat of a spurious model (they have a number of KRAS pathway mutations which are otherwise rare in TNBC). I would have liked to see a PDX model of TNBC/BLBC, or at least the BT549 model as an alternative therapeutic model.

3) While the data are very thorough, in many cases the effect sizes are quite small, and don't seem to correlate, for example, with degree of shRNA knockdown (see for instance shRNA against RSK3 is weak in terms of downregulation of protein, yet the effect sizes are remarkably consistently greater).

4) While the statistical section lists the designations for *, **, *** etc, they should be in the figure legends. In addition, many of the figure legends are unclear with numerous shorthand notations of drugs etc, without identification in the legend. For instance, in figure 2 ChIP experiments, what is SP and NSP? Overall I felt the legends lack significant detail and need to be more thoroughly written.

5) My major issue with the work (if there was one) is that overall there is very thorough and detailed mechanism, but the practical upshot - that EGFR inhibitors combined with BET inhibitors seems to be an effective combination is underdeveloped for the scope of the Journal. I would have preferred to see at least one or two non-MDA-231 tumor models, as mentioned before, this is not a good predictive model (there have been literally hundreds of effective combinations shown in MDA-231 models that have not panned out in the clinic).

Minor points:

1) The authors should be careful to remove uses of the words prove/d/en, as this is highly arguable in most cases.

2) Figure 3A is gratuitous (unless the individual data sets are pure luminal A, B, Her2, BLBC, etc). They are not defined in the legend as to their composition, there are 8 different relatively small datasets, and if they are mixed breast cancer tumors, then Figure 3B offers an obvious confounder in the analysis in that the correlations may be only due to differences between luminal and basal phenotypes. The authors should endeavor to simply use the TCGA, organize by subtype, and then draw correlations of the two parameters in only luminal A, only luminal B, only BLBC, etc.

Point-By-point Response to the Reviewers' Comments

We thank for the helpful and constructive comments from reviewers and appreciate the opportunity for us to revise the work. We have taken the comments from all reviewers seriously and revised our manuscript extensively. Below, a series of data have been supplied to support the points.

New data and changes

1. Use of two additional BET inhibitors, iBET151 and iBET762, to see the combined killing effect of BET/EGFR inhibition (**New Figure 6B**) and the induction of JunD/RSK3 signaling (**Figure S6A**).
2. Use of MDA-MB-468 cell line as additional BLBC model to observe the combined killing effect of BET/EGFR inhibition which detected by *in vitro* cell growth assay (**Figure S6E**) and *in vivo* xenograft mice model (**Figure S6F-H**).
3. Re-analysis of the correlation of expression of *JunD* and *RSK3* genes in relatively pure datasets of breast cancer (**New Figure 3A**).
4. Employment of ChIP assay to see the binding status of BRD2 and BRD3 on the promoter of miRNA (**Figure S4J**).
5. Use of ChIP assay to observe the association of c-Jun, JunB and c-Fos on the enhancer region of *RSK3* gene (**Figure S2C**).
6. Performance of ChIP assay to detect the distribution of H3K4me1, H3K4me3 and H3K27ac on the binding region of JunD (**Figure S2A**).
7. Measurement of the effect of p38 inhibitor SB203580 in combination with JQ1 which detected by CCK8 growth assay (**Figure S5B**).
8. Use of EMSA assay to observe the binding on the target region (**Figure S2B**).
9. Modification of the manuscript and supplement of necessary illustrations in the Figure legends. The changes we have made are highlighted with red color in the manuscript.

Reviewer #1 (Remarks to the Author; expert in BET inhibitors resistance:

The authors investigate the mechanism of JQ1 resistance in the breast cancer. They found that JQ1 treatment evicts BRD4 from the Foxd3 localized miRNA promoter and repress the expression of miR-548d-3p. Furthermore, they showed that miRNA loss causes the restored JunD and RSK3. Dual inhibition of ERK or EGFR inhibition overcomes the BET inhibitor resistance. Findings are very novel, interesting and mechanistic. However, there are a few issues need to be addressed.

Response: Thank for the positive comments and appreciate the opportunity for us to revise our work.

Major points:

1. *In the paper, the authors defined a mechanism of JQ1 resistance in breast cancer cells. How about other BET inhibitors, like iBET151 and iBET762. The authors should repeat some key experiments using the other iBET inhibitors.*

Response: We appreciate the great suggestions from Reviewer #1 and have performed several experiments to observe the synergistic effects of Osmeritinib on iBET151 or iBET762. Results showed that Osmeritinib produces significantly synergistic effects on these two inhibitors which detected by cell proliferation assay (**New Figure 6B**). Also, both iBET151 and iBET762 induce obvious JunD/RSK3 expression in BLBC cells (**Figure S6A**). These data indicate that double BET/EGFR inhibition is an effective way to kill BET-resistant breast cancer cells.

2. *The authors found BRD4 can bind to miRNA's promoter. How about BRD2 and BRD3? Do they bind to the same promoter? Does JQ1 or iBET also block the BRD2 and BRD3 binding in this promoter?*

Response: Thank for the constructive suggestions. Following the construction, we observed the binding status of BRD2 and BRD3 on the miRNA promoter by ChIP assay. Results displayed that these two proteins cannot associate with the miRNA gene promoter (**Figure S4J**), suggesting BRD4 is the only member of BET family which regulates the expression of miRNA-548d-3p.

3. *The authors found that JunD can bind to the promoter of RSK3. JunD is the member of AP-1. AP-1 also has other members, including Fos, c-Jun and JunB, they share the similar binding motif (Hernandez, J.M.et al, Oncogene 2008). The authors should rule out other members' function in the RSK3 promoter.*

Response: We appreciate the insightful comment from Reviewer #1. We have performed the corresponding ChIP experiments using specific antibodies in the absence or presence of JQ1 treatment. Results showed that although these four proteins all recognize the RSK3 enhancer in the absence of JQ1 treatment, c-Jun and JunD have the stronger binding affinity, while JunB and c-Fos show much weaker association. Upon JQ1 treatment, the binding of c-Jun is significantly decreased; although the binding of JunB and c-Fos is slightly elevated, however, the binding affinity of JunD on RSK3 enhancer is robustly increased in the presence of JQ1 (**Figure S2C**). Taken together, we reason that JunD is most likely to determine the responsive RSK3 expression and BET resistance.

4. Also for the *RSK3* promoter, the -1771 bp is not in the region of promoter, the promoter region is -800 - +200bp (Kwon et al, PNAS, 2007). It should be an enhancer region or the authors should show the H3K4me1 and H3K4me3 data in the breast cancer cells.

Response: We appreciate the illuminating comment from Reviewer #1; we have performed the ChIP experiments using antibodies of H3K4me1 and H3K4me3. Results showed that there are relatively strong signals of H3K4me1 on the binding region of JunD, while the modification of H3K4me3 is quite weak (**Figure S2A**), suggesting that JunD recognizes the active enhancer region of *RSK3* gene. We have corrected our expression into ‘by searching enhancer region of *RSK3* gene, we found a potential JunD binding site, GTGACTCT (-2,161 bp upstream of the translation start site)’ in the manuscript.

5. In Figure 2E, the ChIP assay cannot demonstrate the direct binding for proteins and DNAs. The authors should test the direct binding using EMSA.

Response: Thank for the constructive suggestions; we have detected the binding using LightShift™ Chemiluminescent EMSA Kit (Thermo Fisher, #20148), the biotin labelled DNA probes are 5'-GTCTTTTATGAGTGACTCTCCTAGCTTTTT-3' and 5'- AAAAAGCTAGGAGAGTCACTCATAAAAGAC-3' (**Figure S2B**).

6. In Figure 3C and 3D, MCF-7 is not a HER2+ cell line (K Subik et al., 2010).

Response: Thank for the constructive suggestions; we have corrected it.

7. In Figure S5, the authors should rule out the other MAPK members, like JNK and p38.

Response: Thank for the constructive suggestions; the synergistic effect of JNKs inhibitor JNK-IN-8 has been found to be weaker than that of silencing of *RSK3* (**Figure S5A**). P38 inhibitor SB203580 has no significant capacity to enhance the killing effect of JQ1 (**Figure S5B**).

Minor points:

1. In figures 3A and S3A, some of the labeled words are too small to be seen clearly.

Response: Thank for the constructive suggestions; we have replaced these charts with photos labelling with larger words.

Reviewer #2 (Remarks to the Author); expert in breast cancer signalling:

Tai et al present a nice molecular report in which they delineate a pathway explaining

some of the preferential activity of BET inhibitors (specifically JQ1) toward TNBC/BLBC. The authors use two TNBC cell lines primarily to demonstrate this effect: MDA-231 and BT549. They demonstrate very thoroughly a mechanism whereby FoxD3 and BRD4 interactions are disrupted by JQ1 leading to reduced transcription of MiR548D1, leading to loss of JunD regulation. JunD-mediated transcription of RSK3 drives therapeutic resistance, apparently requiring upstream activation of ERK1/2/5, which can be blocked by broad-spectrum MEK inhibition or EGFR inhibition (at least in MDA-231). The data are quite thoroughly done and include both chemical and genetic inhibition, although the latter is inferior (weakly repressive shRNAs complicate the analyses). Although I would not request the authors to repeat the entire work using something more clean (better shRNAs or CRISPR), I would suggest they use better genetic inhibitors in the future. Overall the data presented support a nice complete mechanism, although the therapeutic upshot with more thorough and robust *in vivo* models is a missed opportunity.

Response: Thank for the positive comments and appreciate the chance for us to revise our work.

Major points:

1) At least a few of the linear mechanisms should be shown with a different BET inhibitor (unless I missed it) to confirm this is a class-specific effect

Response: We appreciate the great suggestions from Reviewer #2, we have performed several experiments to observe the synergistic effects of Osmeritinib on iBET151 or iBET762, two other commercial BET inhibitors; results showed that Osmeritinib also produces significantly synergistic effects on these two inhibitors which detected by cell proliferation assay (**New Figure 6B**). Also, both iBET151 and iBET762 induce obvious JunD/RSK3 expression in BLBC cells (**Figure S6A**). These data indicate that double BET/EGFR inhibition is an effective way to kill BET-resistant breast cancer cells.

2) The only tumor model used in the manuscript in MDA-231, which, while widely used as a 'model' BLBC cell line, most breast cancer researchers agree that they are somewhat of a spurious model (they have a number of KRAS pathway mutations which are otherwise rare in TNBC). I would have liked to see a PDX model of TNBC/BLBC, or at least the BT549 model as an alternative therapeutic model.

Response: We appreciate the illuminating suggestions from Reviewer #2. Since BT549 is difficult to grow in nude mice (we have tried the experiment but failed), then we used another typical BLBC cell line MDA-MB-468 as our xenograft model to detect the combined killing effect of BET/EGFR inhibition. We obtained similar results from *in vitro* CCK8 proliferation assay (**Figure S6E**) and nude mice xenograft

model (**Figure S6F-H**). These data further indicate the effectiveness of double BET/EGFR inhibition in treatment of BLBC.

3) *While the data are very thorough, in many cases the effect sizes are quite small, and don't seem to correlate, for example, with degree of shRNA knockdown (see for instance shRNA against RSK3 is weak in terms of downregulation of protein, yet the effect sizes are remarkably consistently greater.*

Response: Thank for the constructive suggestions. We analyzed the efficiency of RSK3-knockdown in our western blot results (Figure S1E) by Image-J software, showing that the protein levels of RSK3 in shRNA-clones are decreasing to nearly 40% of control. Although the knockdown efficiency is not very high, the greater subsequent inhibition-effect should be due to the synergistic effect induced by double BRD4/RSK3 inhibition based on our data. Probably, the cells with highest knockdown-efficiency cannot live on during the clone selection, because although knockdown of RSK3 itself does not influence the growth of cells, we cannot exclude the possibility that RSK3 is involved in other important biological functions.

4) *While the statistical section lists the designations for *, **, *** etc, they should be in the figure legends. In addition, many of the figure legends are unclear with numerous shorthand notations of drugs etc, without identification in the legend. For instance, in figure 2 ChIP experiments, what is SP and NSP? Overall I felt the legends lack significant detail and need to be more thoroughly written.*

Response: Thank for the constructive suggestions; we have carefully revised the manuscript: listed the designation of ‘*, **, ***’ and ‘SP and NSP’ in the figure legends; expanded the illustration of figure legends.

5) *My major issue with the work (if there was one) is that overall there is very thorough and detailed mechanism, but the practical upshot - that EGFR inhibitors combined with BET inhibitors seems to be an effective combination is underdeveloped for the scope of the Journal. I would have preferred to see at least one or two non-MDA-231 tumor models, as mentioned before, this is not a good predictive model (there have been literally hundreds of effective combinations shown in MDA-231 models that have not panned out in the clinic.*

Response: We appreciate the illuminating suggestions from Reviewer #2; we have used another typical BLBC cell line MDA-MB-468 as our xenograft model to detect the combined killing effect of BET/EGFR inhibition. We obtained similar results from *in vitro* CCK8 proliferation assay (**Figure S6E**) and nude mice xenograft model (**Figure S6F-H**). These data further indicate the effectiveness of double BET/EGFR inhibition in treatment of BLBC.

Minor points:

1) *The authors should be careful to remove uses of the words prove/d/en, as this is highly arguable in most cases.*

Response: We thank the constructive suggestion; we have carefully removed all these kinds of words or statements.

2) *Figure 3A is gratuitous (unless the individual data sets are pure luminal A, B, Her2, BLBC, etc). They are not defined in the legend as to their composition, there are 8 different relatively small datasets, and if they are mixed breast cancer tumors, then Figure 3B offers an obvious confounder in the analysis in that the correlations may be only due to differences between luminal and basal phenotypes. The authors should endeavor to simply use the TCGA, organize by subtype, and then draw correlations of the two parameters in only luminal A, only luminal B, only BLBC, etc.*

Response: We appreciate the illuminating suggestions from Reviewer #2; we have revised and updated these analyses, specifically, we respectively analyzed the expression levels of JunD and RSK3 in four datasets from GEO (Gene Expression Omnibus): GSE76275 (was separated into two sub-groups: Triple-negative and non-Triple-negative); GSE43358 (was separated into two sub-groups: Triple-negative and non-Triple-negative); GSE42568 (was separated into two sub-groups: ER-positive and ER-negative); GSE31448 (was separated into two sub-groups: Luminal A and Luminal B). Based on these relatively pure datasets, we also found the similar results which the expression of JunD and RSK3 is positively correlated in different subtypes of breast cancer (**New Figure 3A**).

REVIEWERS' COMMENTS:

Reviewer #1 (Remarks to the Author):

The authors have addressed all my concerns. The current version is suitable for publication in Nature Communications.

Reviewer #2 (Remarks to the Author):

My concerns have all been addressed - great work!

REVIEWERS' COMMENTS:

Reviewer #1 (Remarks to the Author):

The authors have addressed all my concerns. The current version is suitable for publication in Nature Communications.

Response: Thank for the positive comments from Reviewer #1.

Reviewer #2 (Remarks to the Author):

My concerns have all been addressed - great work!

Response: Thank for the positive comments from Reviewer #2.